# Deep-sea stylasterid $\delta^{18}$O and $\delta^{13}$C maps inform sampling scheme for paleotemperature reconstructions

Theresa M. King[1], Brad E. Rosenheim[1], and Noel P. James[2]

[1]College of Marine Science, University of South Florida, St. Petersburg, 33701, USA
[2]Department of Geological Sciences and Geological Engineering, Queens University, Kingston, K7L 3N6, Canada

*Correspondence to*: Theresa M. King (theresaking@usf.edu)

**Abstract.** Deep-sea corals have the potential to provide high-resolution paleotemperature records to evaluate oceanographic changes in settings that are vulnerable to current and future ocean warming. The isotopic records preserved in coral skeletal carbonate, however, are limited by their large offsets from isotopic equilibrium with seawater. These "vital effects" are the result of biological influences (kinetic and metabolic) on the calcification of coral skeletons and are well known to drive oxygen and carbon stable isotope ratios ($\delta^{18}$O and $\delta^{13}$C, respectively) away from isotopic equilibrium with environmental variables. In this study, two stylasterid corals (*Errina fissurata*) are sampled via cross sections through their primary growth axes to create skeletal $\delta^{18}$O and $\delta^{13}$C maps. Such maps reveal a consistent trend of increasing isotopic values toward the innermost portion of the coral slices; the average center values being ~1 ‰ and ~ 3 ‰ closer to seawater $\delta^{18}$O and $\delta^{13}$C equilibrium values (respectively) than a traditional bulk sample. We investigate possible mechanisms for these unique isotopic trends, including changes in the proportions of aragonite and calcite in these mixed mineralogy corals, and potential growth patterns that would drive spatial isotopic trends. These results highlight the diversity of the stylasterid coral family and the need for additional work to establish $\delta^{18}$O paleotemperature calibrations for deep-sea corals of mixed mineralogy instead of purely aragonite or calcite. Despite the absence of a specific temperature calibration, we can prescribe a sampling scheme for *E. fissurata* corals to achieve accurate paleotemperature reconstructions.

## 1 Introduction

Robust paleoceanographic temperature proxies are fundamental to understanding past climate changes and sensitivities. Foundational work developed the theory and application of the oxygen isotope paleothermometer, which was applied to marine biogenic carbonates including foraminiferal tests and shallow water corals (Urey, 1947; McCrea, 1950; Epstein et al., 1953; Emiliani, 1955; Shackleton, 1967; Emiliani et al., 1978). All archives have limits in geographic distribution despite their ability to lengthen the time domain of paleoceanographic records. On the Antarctic margin, for instance, foraminifera are not widely present in marine sediment cores. Scleractinian zooxanthellate corals are limited to lower latitudes. Deep-sea azooxanthellate corals have been used to elucidate ocean history on different time scales (Adkins et al., 1998; Robinson et al., 2005; Robinson

and van de Flierdt, 2009; Burke and Robinson, 2012; Chen et al., 2020), but normally the extraction of continuous records from individual colonies has been precluded by the complexity of growth habit and the fidelity of elemental and isotopic records archived in their skeletons (Weber, 1973; Wisshak et al., 2009; Robinson et al., 2014). Corals can incorporate high resolution geochemical records over decades to millennia while remaining fixed to the seafloor; this is especially useful to observe regional and global processes causing ocean water masses to heave and shoal at timescales over which the coral is

alive (Andrews et al., 2002; Druffel et al., 1990; Druffel, 1997; Griffin and Druffel, 1989; Risk et al., 2002). Such archives are crucial for paleotemperature reconstructions.

Biomineralization of carbonate coral skeletons records both environmental information and biological effects (also known as "vital effects"), the latter of which must be understood to obtain high-fidelity records of ocean change. Vital effects can obscure environmental information stored in skeletal records as oxygen and carbon stable isotope ratios ($\delta^{18}O$ and $\delta^{13}C$, respectively).

Slow rates of calcification allow for carbon and oxygen isotopes of solid carbonate to approach isotopic equilibrium between skeleton and seawater, a state governed by thermodynamics (McConnaughey, 1989a). Biological calcification, however, includes nonequilibrium fractionation which cannot be interpreted directly as environmental signal (Weber and Woodhead, 1970). Early research on corals demonstrated that isotopic variability can be caused by metabolic fractionation, kinetic fractionation, or a combination of both (McConnaughey, 1989a). The metabolic fractionation is characterized by a change in

carbon isotope composition of the dissolved inorganic carbon pool from which the coral calcifies via incorporation of the products of respiration and photosynthesis of algal symbionts or respiration of the coral itself (Swart, 1983; McConnaughey, 1989a). The kinetic fractionation is described as a product of the kinetic isotope effect: discrimination against heavy oxygen and carbon isotopes during hydration and hydroxylation of $CO_2$ during biomineralization (McConnaughey, 1989b). Rapid calcification results in greater disequilibrium of skeletal $\delta^{18}O$ and $\delta^{13}C$ as the $CO_2$ does not have sufficient time to equilibrate

with ambient seawater before being incorporated into the skeleton (McConnaughey, 1989b).

Vital effects have been invoked to explain skeletal $\delta^{18}O$ and $\delta^{13}C$ values lower than equilibrium, and the strong linear relationship between $\delta^{18}O$ and $\delta^{13}C$ values has been used to further understand vital effects exhibited by corals from varied ocean depths and latitudes (Smith et al., 2000; Emiliani et al., 1978; Heikoop et al., 2000; Mikkelsen et al., 2008; McConnaughey, 1989a). Early work by Emiliani et al. (1978) examined the $\delta^{18}O$ and $\delta^{13}C$ recorded by a solitary scleractinian

coral (class Hexacorallia, order Scleractinia) and found that both isotopic ratios trended toward higher values from the bottom to top of the coral. This was interpreted as a slowing growth rate with time, approaching isotopic equilibrium with the surrounding seawater (Emiliani et al., 1978). Additional work on solitary scleractinians by Adkins et al. (2003) employed a microsampling approach for $\delta^{18}O$ and $\delta^{13}C$ that identified new mechanisms for vital effects. Along with the lowest isotopic values occurring at regions of rapid calcification, Adkins et al. (2003) observed a break in the linear relationship of stable

isotopic ratios in these regions. The authors hypothesized that rapid biomineralization drives the internal calcifying fluid pH toward higher values, increasing the pumping of $CO_2$ inward, which stabilizes fractionation of carbon isotopes, yet further

fractionates oxygen isotopes due to pH driven changes in carbonate speciation (Adkins et al., 2003). Later work on bamboo corals (class Octocorallia, order Malacalcyonacea) sampled across and along their vertical growth axes resulted in low $\delta^{18}O$ and $\delta^{13}C$ values near the innermost portion of the coral and at the distal tips (Hill et al., 2011). Interpreting the lower isotope ratios, however, was complicated because the assumed faster calcification rates in these regions were not supported by calculated growth rates, and the locations of maximum growth rates were not consistent for a single specimen (Hill et al., 2011). A further understanding of vital effects was established by Chen et al. (2018) who investigated the role of the carbonic anhydrase enzyme which catalyzes the hydration and hydroxylation of $CO_2$. The authors determined that the amount of the enzyme in the calcifying fluid determines the internal speciation of carbonate and thus, alters the slope of the linear $\delta^{18}O$ and $\delta^{13}C$ relationship (Chen et al., 2018).

The need for accurate paleotemperature archives underscores the need to understand the impact of vital effects on deep-sea corals. Stylasterid corals (class Hydrozoa, order Anthoathecata) are a ubiquitous taxon of deep-sea corals that occupy ocean depths from the surface to greater than 2700 m, and can be found in high and low latitudes, having great potential for paleoceanographic reconstructions (Cairns, 2011). Stylasterid geochemical work has expanded over the last several years, but because of the inherent diversity of this coral family, research is still needed to understand vital effect-induced isotope fractionation. Early work by Weber and Woodhead (1972) demonstrated that shallow water stylasterids (genus *Distichopora* and *Stylaster*) had precipitated their skeletons closer to isotopic equilibrium than some scleractinian corals. Later work by both Wisshak et al. (2009) and Black and Andrus (2012) examined possible diagenetic effects on the stable isotopic geochemistry of *Errina dabneyi* and *Stylaster erubescens*, respectively. More recently, Samperiz et al. (2020) have produced a compilation of stylasterid skeletal $\delta^{18}O$ and $\delta^{13}C$ records from nearly 100 specimens. Similar to the deep-sea scleractinian corals, Samperiz et al. (2020) found the lowest $\delta^{18}O$ and $\delta^{13}C$ values in the innermost portion of the main coral trunk and at the distal growth tips, supporting rapid growth in those regions. This work also found that skeletal mineralogy influences $\delta^{18}O$ and $\delta^{13}C$ values, whereby calcitic stylasterids record slightly lower values than their aragonitic counterparts (Samperiz et al., 2020). Stylasterid carbonate skeletons can consist of aragonite, calcite, or a mixture of both polymorphs (Cairns and Macintyre, 1992; Kershaw et al., 2023). Another characteristic thus far unique to stylasterids is the absence of pH upregulation within the calcifying fluid (Stewart et al., 2022). The analysis of boron isotopes from calcitic, aragonitic, and mixed stylasterids (*Errina* spp and *Stylaster* spp) compared to those of scleractinian corals support a completely different calcification method wherein stylasterids do not elevate internal pH to promote calcification (Stewart et al., 2022). A largely unknown calcification strategy, paired with variable mineralogy that could impact stable isotopic records highlight a need to develop a deeper understanding of these corals so that we may accurately reconstruct paleoceanographic conditions.

Here we establish a link between isotopic changes and growth pattern of a deep-sea stylasterid using two *Errina fissurata* specimens. We generate $\delta^{18}O$ and $\delta^{13}C$ maps over coral surfaces perpendicular to and along vertical growth axes to evaluate the locations and magnitudes of isotopic variability. We compare our skeletal maps to calculated seawater equilibrium values,

mineralogical data, and isotopic trends modelled from hypothesized coral growth scenarios to determine the influences on
$\delta^{18}O$ and $\delta^{13}C$ and potential implications for paleoceanographic reconstructions. Contradictory to previous research, the isotopic values most representative of environmental signals in these corals are recorded in the innermost portion of the coral stems. Ultimately, we prescribe targeted sampling, avoiding the use of bulk drilling methods for the most accurate paleotemperature reconstructions for *E. fissurata*.

## 2 Methods

### 2.1 Study location and specimen collection

The Ross Sea is a region of bottom water formation for the worlds' oceans, a characteristic that influences the local oceanography in which the stylasterids live. The region experiences seasonal katabatic winds that create
sea ice-forming polynyas, which in turn, create High Salinity Shelf Water through the process of brine rejection (Kurtz and Bromwich, 1985; Picco et al., 2000). The High Salinity Shelf Water flows along the western Ross Sea and out to the shelf edge where it
mixes with upwelled modified Circumpolar Deep Water, resulting in a component of dense Antarctic Bottom Water that spills down the continental slope (Gordon et al., 2009; Jacobs et al., 1970; Sandrini et al., 2007).

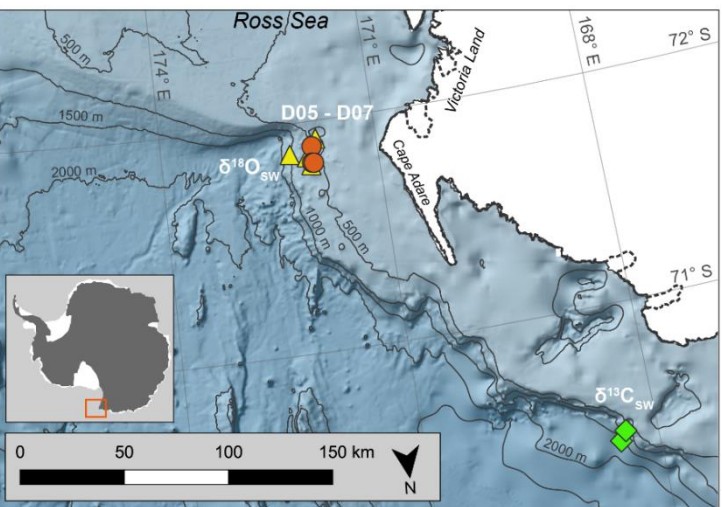

**Figure 1:** Map of coral collection sites. D05–D07 represent dredge sites during NBP07-01 from which coral samples were collected on the outer western Ross Sea continental shelf. The diamonds and triangles mark the stations used to calculate the seawater isotopic ratios ($\delta^{13}C_{sw}$ and $\delta^{18}O_{sw}$; Table S2 in the Supplement). Antarctic landmass and ice sheets are colored white (Gerrish et al., 2022) with the coastline marked by a solid black line, and ice sheet grounding line denoted by a dotted black line (Mouginot et al., 2017). Bathymetric contours are in increments of 500 m (Arndt et al., 2013). Antarctic inset denotes sampling site marked by red box. Map was created using the Quantarctica data set collection within QGIS (Matsuoka et al., 2021).

The stylasterid coral specimens for this study were collected aboard the U.S. Antarctic Program expedition NBP07-01 near Cape Adare in the western Ross Sea. Seamounts on the outer continental shelf were dredged at a water depth ranging 400 m to 600 m and the corals
used here were from the fifth through seventh dredges (D05–D07; Fig. 1, Table 1). The recovered stylasterid corals were predominantly *Errina* spp., most likely *Errina fissurata* based on morphological descriptions (Cairns, 1983a, b, 1991) and scanning electron microscopy (SEM). Specimens were recovered both alive and dead (as evidenced by their pigmentation), some with growth tips intact. For this study, one live-collected and one dead-collected stylasterid were selected for isotopic analyses (EA-11 and EA-12, respectively), targeting the
longest whole specimens ranging from ~9 to 10 cm long (Fig. 2). For SEM analyses, different corals from the same dredge

were analyzed as they were sputter-coated in gold palladium and not suitable for geochemical analysis. Specimens EA-20 and EA-21 were used for taxonomical identification and EA-22 through EA-24 for physical evidence of diagenesis (e.g., Wisshak et al., 2009).

## 2.2 Coral sampling and isotope analysis

The stylasterid specimens were sampled for stable oxygen and carbon isotope measurements ($\delta^{18}O$ and $\delta^{13}C$, respectively) over cross sections of their 135 major growth axes. A Gryphon diamond band saw was used to slice discs measuring approximately 2–3 mm thick from each specimen's main 140 stems (three from EA-11 and

| Lat. (°S) | Long. (°E) | Water Depth (m) | Dredge | Temp. at Depth (°C) | $\delta^{18}O$ seawater (‰, SMOW) | $\delta^{13}C$ seawater (‰, PDB) |
|---|---|---|---|---|---|---|
| 71.89 | 171.9 | 490-593 | D05 | $-0.10 \pm 0.09$ | $-0.26 \pm 0.06$ | $0.66 \pm 0.05$ |
| 71.82 | 171.92 | 518-643 | D06 | | | |
| 71.82 | 171.9 | 489-599 | D07 | | | |

**Table 1:** Sample collection data including approximate coordinates of dredges, the range of water depths sampled, names of each dredge, and approximate seawater properties. Temperature is averaged for dredge depth range using measurements from nearby AnSlope stations (Visbeck, 2015; Jacobs, 2015; Gordon, 2016). The corresponding potential temperatures and salinities for this depth range were used to determine the $\delta^{18}O$ of the seawater based on values reported by Jacobs et al. (1985) for Antarctic margin water masses. The seawater $\delta^{13}C$ is an average of values reported for the same depth range from nearby World Ocean Circulation Experiment stations (WOCE, 2002; Fig. S1 in the

two from EA-12; Fig. 2). Additionally, a longer 8 mm thick section was cut from the lower main stem of EA-11 (EA-11d; Fig. 2). This disc was sliced in half lengthwise with the band saw and sampled along the vertical face. Each coral disc was sonicated in DI water until no remaining loose particles were released then dried in a 50° C oven for 24 hours. The coral discs were then drilled over their surfaces with a New Wave MicroMill system at grid spacing that varied between 1 and 1.5 mm, based on the

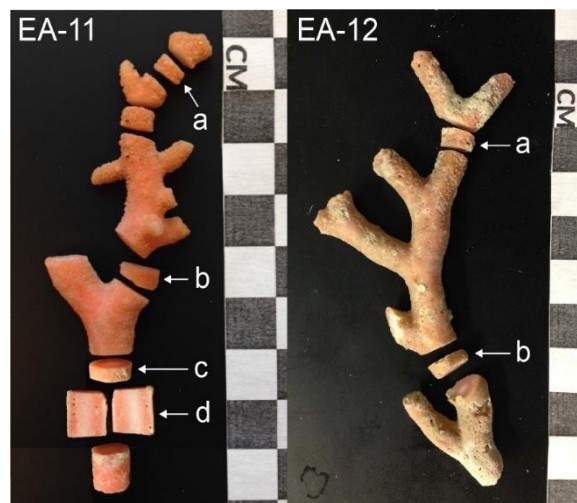

**Figure 2:** Whole coral specimens EA-11 (left) and EA-12 (right). EA-11 was live-collected, and EA-12 was dead. Sample discs are labelled, and the corresponding isotope maps are in Fig. 3.

diameter of the slice and ampullae in the disc. The MicroMill was configured with a 0.5 mm round carbide bur bit for each hole that plunged to a maximum depth of 1 mm, just far enough to obtain enough carbonate material for analysis (~80–120 μg $CaCO_3$). The carbonate powder from each hole was transferred to vials and the sample hole and stylasterid surface were cleaned with compressed air or nitrogen to remove any residual powder between drilling intervals.

The sample vials were flushed with helium and acidified with phosphoric acid at 50° C to generate $CO_2$ that was analyzed at the University of South Florida College of Marine Science using a Thermo Scientific MAT 253 stable isotope ratio mass spectrometer with a Gas Bench II preparatory device. All reported values are in standard delta ($\delta$) notation and reported as per mil (‰). Laboratory

reference materials Borba ($\delta^{13}$C: 2.89 ‰, $\delta^{18}$O: -6.15 ‰), TSF-1 ($\delta^{13}$C: 1.95 ‰, $\delta^{18}$O: -2.20 ‰), and Leco ($\delta^{13}$C: -15.44 ‰, $\delta^{18}$O: -20.68 ‰), were used for instrument correction and normalizing to the Pee Dee Belemnite scale (PDB), and an internal

Antarctic coral standard was used for quality control. To account for fractionation of oxygen isotopes during acidification (Kim et al., 2007) we measured calcite reference materials, which corrected for fractionation during the transfer onto the PDB scale. The analytical uncertainty (1$\sigma$) of the MAT 253 during this study was ±0.083 ‰ $\delta^{13}$C and ±0.064 ‰ $\delta^{18}$O.

### 2.3 Mineralogical analysis

Coral slice EA-11d was analyzed for mineralogical analysis using X-ray diffraction (XRD) after it was sampled for stable

isotope analysis. The flat surface was not powdered, but rather scanned to isolate each vertical stripe across the pink, white, and pink colors (Fig. 3). The diffraction data were collected on a Bruker D8 Advance Powder Diffractometer with Lynxeye detector and motorized slits assembly at the University of South Florida X-ray Diffraction Facility and Solid State Characterization Core Lab. The coral slice was centered within a beam of very small divergence, 0.02°, that was able to probe a single band of color at a time. The crystalline phases were identified with Bruker-EVA 7 software and the Crystallographic

Open Database (COD). Two additional *E. fissurata* specimens were sampled for mineralogical analysis via XRD, one live-collected and one dead-collected (EA-13 and EA-14, respectively). Two samples were collected from each specimen, one in the center, white region and the other in the mid to outer pink region. The samples were ground, mixed with a quartz standard, and smeared on a glass slide. They were analyzed on a Malvern Panalytical Empyrean Multipurpose Diffractometer at Queen's University Facility for Isotope Research, Ontario, Canada. Original total mineralogy was ascertained by running from 20.0–

45.0 2$\theta$. Detailed mineralogy was determined by running from 34.0–35.5 2$\theta$. The reported relative mineralogical amounts are semi-quantitative.

### 3 Results

The interior coral skeletons lacked any visible banded growth structure

that we expected to see in the manner of Samperiz et al. (2020) and Wisshak et al. (2009). Instead, the corals were characterized by a central white section, surrounded by shades of pink

in the mid to outer sections (Figs. 2 and 3). The replacement of a high-density growth banding structure with the observed color blocking is an obstacle to estimating the amount of

| Coral Slice | $\delta^{18}$O (‰, PDB) | | $\delta^{13}$C (‰, PDB) | | |
| --- | --- | --- | --- | --- | --- |
| | Maximum | Minimum | Maximum | Minimum | n |
| EA-11a | 2.48 | 0.65 | -3.72 | -9.56 | 17 |
| EA-11b | 2.18 | 0.87 | -4.34 | -8.61 | 24 |
| EA-11c | 2.64 | 1.08 | -2.84 | -8.54 | 20 |
| EA-11d | 2.87 | 0.66 | -2.64 | -8.68 | 37 |
| EA-12a | 2.81 | 0.60 | -3.19 | -8.18 | 17 |
| EA-12b | 2.53 | 0.82 | -3.00 | -8.02 | 25 |
| All | 2.87 | 0.60 | -2.64 | -9.56 | 140 |
| Linear Regression (All data) | Slope | Intercept | $R^2$ | p-value | n |
| | 2.88 (± 0.14) | -10.94 (± 0.22) | 0.76 | < 0.001 | 140 |

**Table 2:** Summary data for *E. fissurata* and regression statistics for the resulting compilation (see Fig. 4).

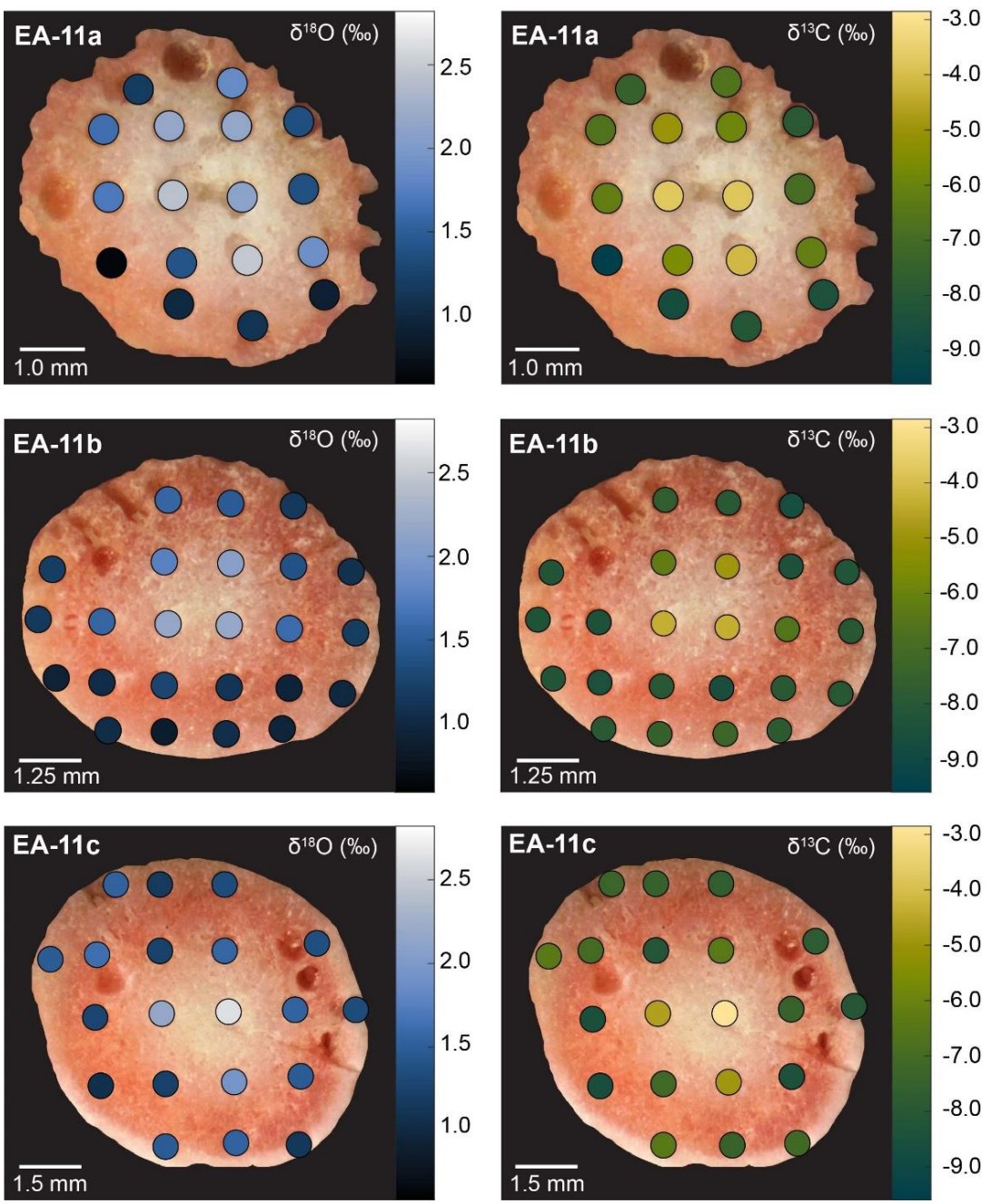

time over which these corals have been growing. Thus, researchers aiming to analyze this coral taxon must employ additional radiometric dating techniques (e.g., Cheng et al., 2000) that are beyond the scope of this work. Although there were no visible banding targets for sampling stable isotopes, the grid spacing of our sampling scheme allowed for microdrilling of several representative samples of both pink and white areas across each coral slice (Fig. S1 in the Supplement).

**Figure 3:** Stable isotope ratio maps for slices from specimens EA-11 and EA-12. Each horizontal pair of images show data from the same coral slice: the left column depicts $\delta^{18}$O values and the right column $\delta^{13}$C values. The measurements are presented as colored circles on each slice with corresponding adjacent color bar. Values are expressed in per mil (‰) relative to PDB, and analytical uncertainty (1σ) during this study was ±0.064 ‰ $\delta^{18}$O and ±0.083 ‰ $\delta^{13}$C. All slices exhibit the same feature of the highest isotopic values toward the inner white section, which is not always the geometric center of the slice.

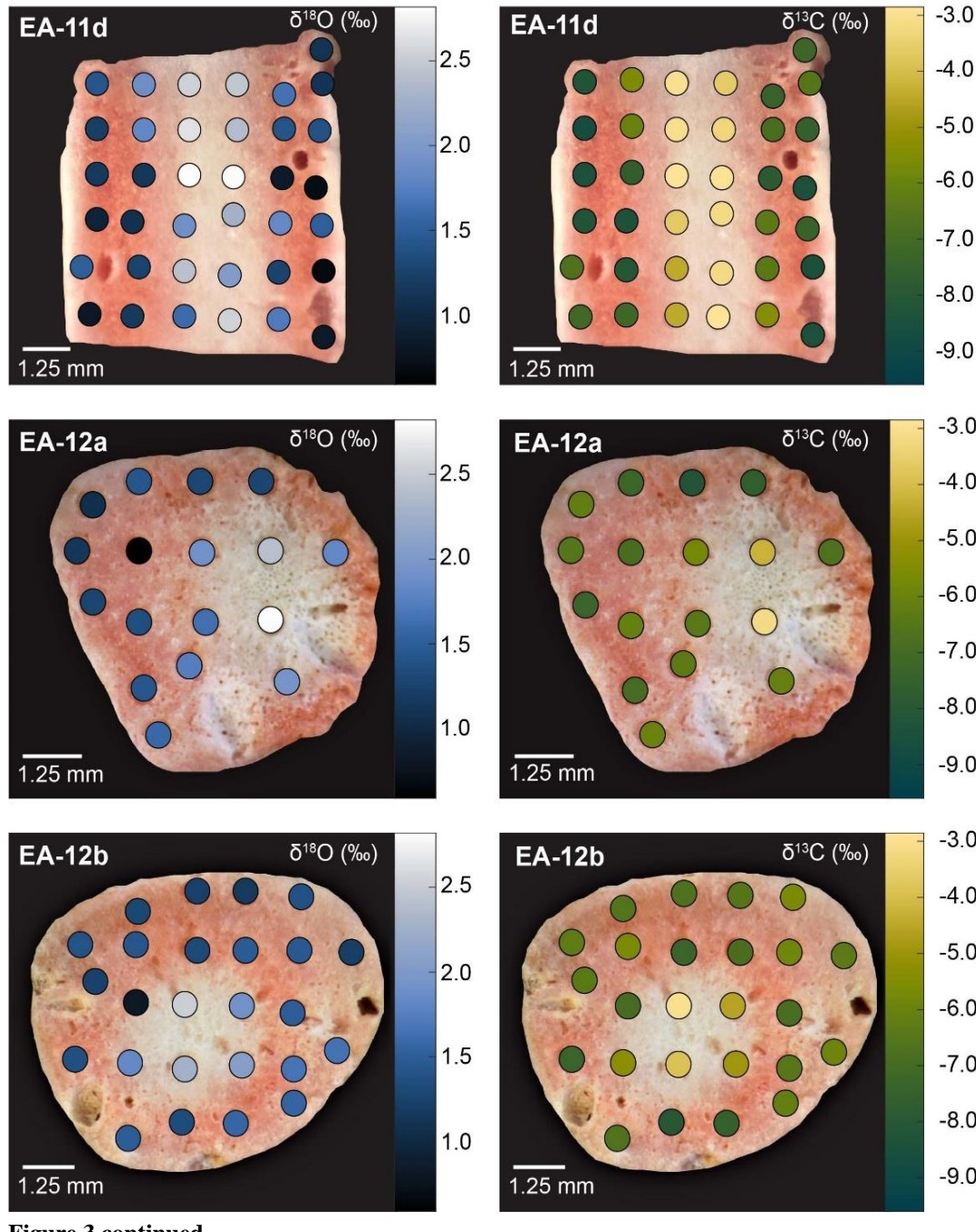

## 3.1 Stable carbon and oxygen isotope trends

Each coral slice exhibits a wide range of stable carbon and oxygen isotope ratios that are higher in the center, white section, and lower in the outer pink section of the coral (Fig. 3; Table S1 in the Supplement). For specimen EA-11, the ranges in $\delta^{18}O$ and $\delta^{13}C$ values across slices EA-11a through EA-11c are ~1.5 ‰ and ~5.2 ‰, respectively. This variability is too large to produce accurate paleoceanographic records without knowledge of where the most accurate, or closest to that of an environmental signal, is preserved in the

**Figure 3 continued.**

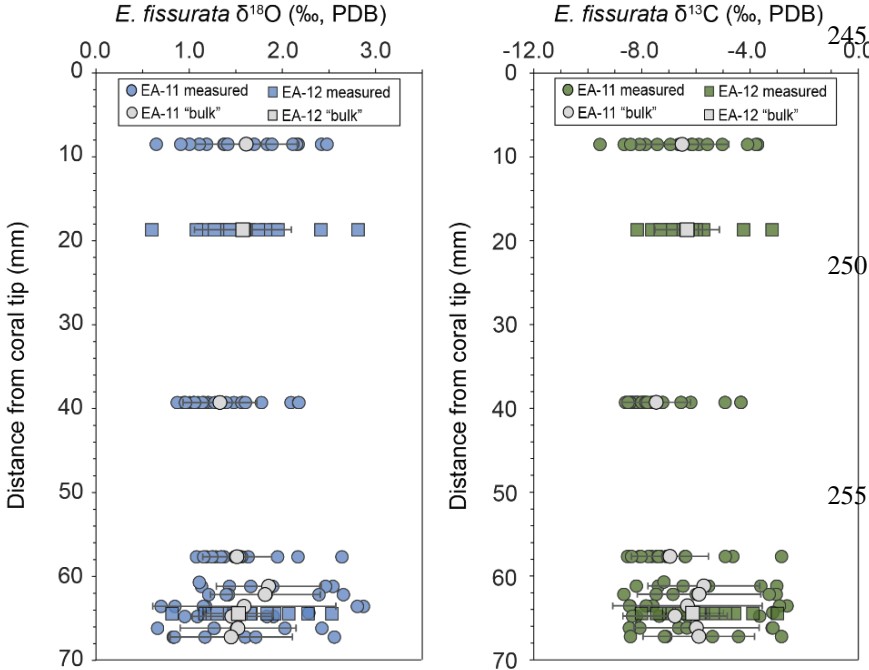

**Figure 4:** Stable isotope ratios from each slice versus distance from the coral tips. On the left is $\delta^{18}O$ and on the right is $\delta^{13}C$; the analytical uncertainty for each is smaller than the data points. A "bulk" value was calculated as the average for each slice that would be similar to common bulk drilling methods (except for EA-11d which was cut transverse to the growth axis and spans a range of distances, each averaged separately. Error bars on "bulk" calculations represent standard deviation for each slice (or distance from the tip). The "bulk" values do not vary along the growth axis.

coral skeleton. The slightly larger range of $\delta^{18}O$ and $\delta^{13}C$ values observed over EA-11d (2.2 ‰ and 6.0 ‰, respectively) is not as surprising as it could be due to the sampling scheme for this slice covering a larger vertical distance of coral than the others (i.e., transecting more time intervals). The compilation of EA-11 $\delta^{18}O$ values ranges from 0.65 ‰ to 2.87 ‰, and the $\delta^{13}C$ values range from -9.56 ‰ to -2.64 ‰, with the minimum values exhibited by EA-11a (nearest the tip) and maximum values by EA-11d (far from the tip; Table 2). Although this trend is not surprising (see Samperiz et al., 2020), if we consider "bulk" values that would be obtained by more traditional coral drilling methods, which average isotopic compositions by drilling into the side of a coral, there is no significant change in either isotopic composition up-coral (Fig. 4).

For specimen EA-12, the range of $\delta^{18}O$ and $\delta^{13}C$ values over each of the slices is on the order of ~2 ‰ and ~5 ‰, similar to EA-11 (Table 2). The entire range of measured EA-12 $\delta^{18}O$ values is 0.60‰ to 2.81‰, and $\delta^{13}C$ values range from -8.18 ‰ to -3.00 ‰. Minimum $\delta^{18}O$ and $\delta^{13}C$ values for EA-12 were both exhibited by slice EA-12a, nearest the tip. The maximum values, however, were not on the same slice (maximum $\delta^{18}O$ at EA-12a and maximum $\delta^{13}C$ at EA-12b). Similar to EA-11, EA-12 also exhibits variability within each slice; however, this also leads to the bulk calculations not being significantly different from each other up-coral (Fig. 4). Looking closer at the individual slices from each coral, the variable stable isotope compositions are significant, i.e., the minimum and maximum values measured on each slice are well beyond the analytical uncertainty (Figs. 4 and 5). This supports that all measurements consistently demonstrate significantly higher $\delta^{18}O$ and $\delta^{13}C$ values in the center of each slice (Figs. 3 and 5).

## 3.2 Deviations from equilibrium with seawater δ¹⁸O and δ¹³C

To determine where the *E. fissurata* skeletons record isotopic compositions closest to an environmental signal, we calculated carbonate isotopic equilibrium values using data reported for nearby hydrographic stations. The seawater stable isotope ratios and temperature were determined using the average composition of the 490–650 m depth range (similar to the coral collection depths; Table S2 in the Supplement) at stations nearest the coral dredge sites (Fig. 1). For seawater δ¹⁸O, we gathered seawater temperature, potential temperature, and salinity records from CTD cast data from the AnSlope (Cross-slope exchanges at the Antarctic Slope Front) project that sampled very close to our dredge sites (Fig. 1 and Fig. S2 in the Supplement) (Jacobs, 2015; Visbeck, 2015; Gordon, 2016). The average potential temperature (-0.12 ± 0.09° C) and salinity (34.7 ± 0.002) were used to identify the water mass at our locale and depth as Low Salinity Bottom Water, as described by Jacobs et al. (1985; Table S2 in the Supplement). The reported δ¹⁸O for this water mass is -0.26 ± 0.06 ‰ (Jacobs et al., 1985).

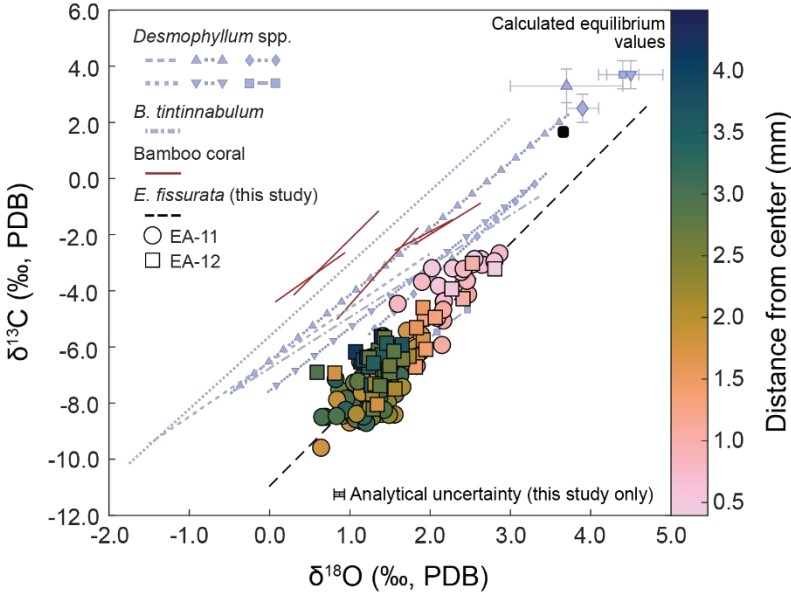

**Figure 5**: Linear regressions of δ¹⁸O vs δ¹³C values for *E. fissurata* compared to aragonitic scleractinian and calcitic bamboo corals. Colors of circles (EA-11) and squares (EA-12) correspond to distance from the coral center, see color bar at the right. Calculated seawater equilibrium value for *E. fissurata* is also shown as a black rounded square (uncertainty is smaller than the square). The dashed black line represents the line of best fit for the isotopic values measured here ($δ^{13}C = 2.88 (\pm 0.14) * δ^{18}O - 10.94 (\pm 0.22)$). Linear regressions for *Desmophyllum* spp. are reported by Adkins et al. (2003) and include *Desmophyllum* sp. (purple line with squares) and D. dianthus (all other *Desmophyllum* lines). The dashed lines with shapes have corresponding equilibria displayed (matching shape with error bars in upper right corner). The *Bathypsammia tintinnabulum* was reported by Emiliani et al. (1978) and Bamboo coral data are from Hill et al. (2011). Lines for external data are not extrapolated beyond the range of reported δ¹⁸O values. The slope of the linear regression produced in this study is similar to those reported for other deep-sea corals, with a similar decrease in both isotopic ratios from equilibrium. The measured values here that are closest to equilibrium are those toward the center of each coral disc.

The results from XRD analyses support the dominant presence of calcite with small amounts of aragonite in these specimens. The color stripes that were analyzed from EA-11d show only small differences between the white and pink compositions (Fig S3 in the Supplement). The powdered samples of EA-13 and EA-14 yielded similar results of mixed mineralogy with ~70 to 77 % calcite (~30 to 23 % aragonite) in the center white portions and ~75 to 80 % calcite (~25 to 20 % aragonite) in the outer pink regions (Table 3). Although there is evidence for mixed mineralogy here, we calculate seawater equilibrium values relative to calcite as that is the main component. The resulting equilibrium δ¹⁸O value was calculated to be 3.66 ± 0.06 ‰

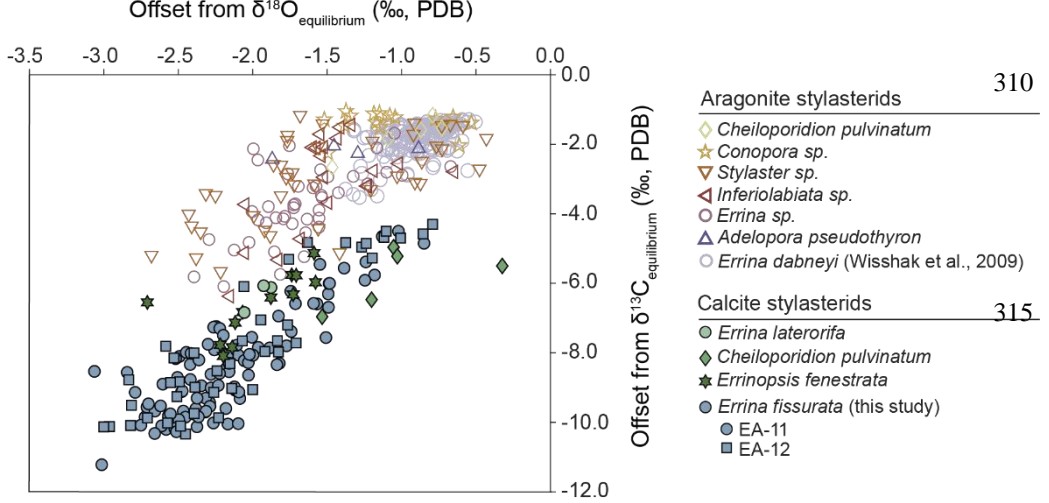

**Figure 6:** Difference between measured coral isotope ratios and equilibrium values for a compilation of stylasterid corals. Except for *E. dabneyi* and *E. fissurata*, all coral isotope data are from Samperiz et al. (2020) and the corresponding equilibrium values were calculated using corresponding seawater data from Samperiz et al. and equations from Grossman and Ku (1986; aragonite δ$^{18}$O equilibrium), O'Neil et al. (1969; calcite δ$^{18}$O equilibrium), and Romanek et al. (1992; aragonite and calcite δ$^{13}$C equilibrium). *E. dabneyi* data and δ$^{18}$O equilibrium are from Wisshak et al. (2009), and the δ$^{13}$C equilibrium was calculated using reported seawater data from Wisshak et al. (2010). The corals from this study (blue filled circles and squares) exhibit a similar offset from equilibrium to other calcitic specimens (filled symbols). Some of the values closer to equilibrium (relative to calcite) from the *E. fissurata*, even overlap with aragonitic samples as well.

using the equation from O'Neil et al. (1969). Determining seawater δ$^{13}$C was straightforward as direct measurements existed for nearby WOCE (World Ocean Circulation Experiment) stations (Fig. 1 and Fig. S2 in the Supplement) (WOCE, 2002). Using the average of reported seawater δ$^{13}$C values for our depth range (0.66 ± 0.05 ‰), we calculated the equilibrium carbonate δ$^{13}$C value as 1.66 ± 0.20 ‰ with the equation from Romanek et al. (1992) (Fig. 5; Table S2 in the Supplement).

Each coral slice exhibits a significant offset from the calculated equilibrium value with the smallest offsets located in the center of each coral slice (Figs. 5 and 7). These equilibrium offsets are all negative in direction (i.e., the corals record isotopic ratios less than expected equilibrium) and range from -3.06 to -0.79 ‰ for δ$^{18}$O and from -11.22 to -4.30 ‰ for δ$^{13}$C (Fig. 6). The offsets from δ$^{13}$C equilibrium are considerably larger than δ$^{18}$O, (Fig. 6). Additionally, we observe a strong linear correlation between δ$^{18}$O and δ$^{13}$C values. The compilation of samples in this study result in a linear trend with a slope of 2.88 ± 0.14 (Δδ$^{13}$C/Δδ$^{18}$O; $R^2$=0.76) or 0.26 ±0.01 (Δδ$^{18}$O/Δδ$^{13}$C; $R^2$=0.76) that passes near the calculated equilibrium value (Fig. 5 and Table 2). Most noticeable is that the values measured from closest to the center of the coral slice are consistently closest to calculated equilibrium, whereas the values from the mid to outer sections are further from equilibrium (Figs. 5 and 7).

## 4 Discussion

### 4.1 Evaluation of major δ¹⁸O and δ¹³C trends

Stable oxygen and carbon isotope analysis of *E. fissurata* supports an influence of vital effects on the skeletal geochemistry of this stylasterid species. Because little is
known about the biocalcification mechanisms and processes of stylasterid corals, we discuss here any similarities to other coral records and present new insights. Fractionation
towards lower isotopic ratios relative to equilibrium as well as a positive correlation between δ¹⁸O and δ¹³C values has been well documented among skeletal records generated
from different classes of corals (e.g., Emiliani et al., 1978; Swart, 1983; McConnaughey, 1989a; Smith et al., 2000; Adkins et al., 2003; Hill et al., 2011; Samperiz et al., 2020; Stewart
et al., 2020). In Figure 5, we compare new data from this study with several scleractinian and bamboo corals. The linear regression between *E. fissurata* δ¹⁸O and δ¹³C values exhibit very
similar slopes to other deep-sea corals and fall within similar ranges of

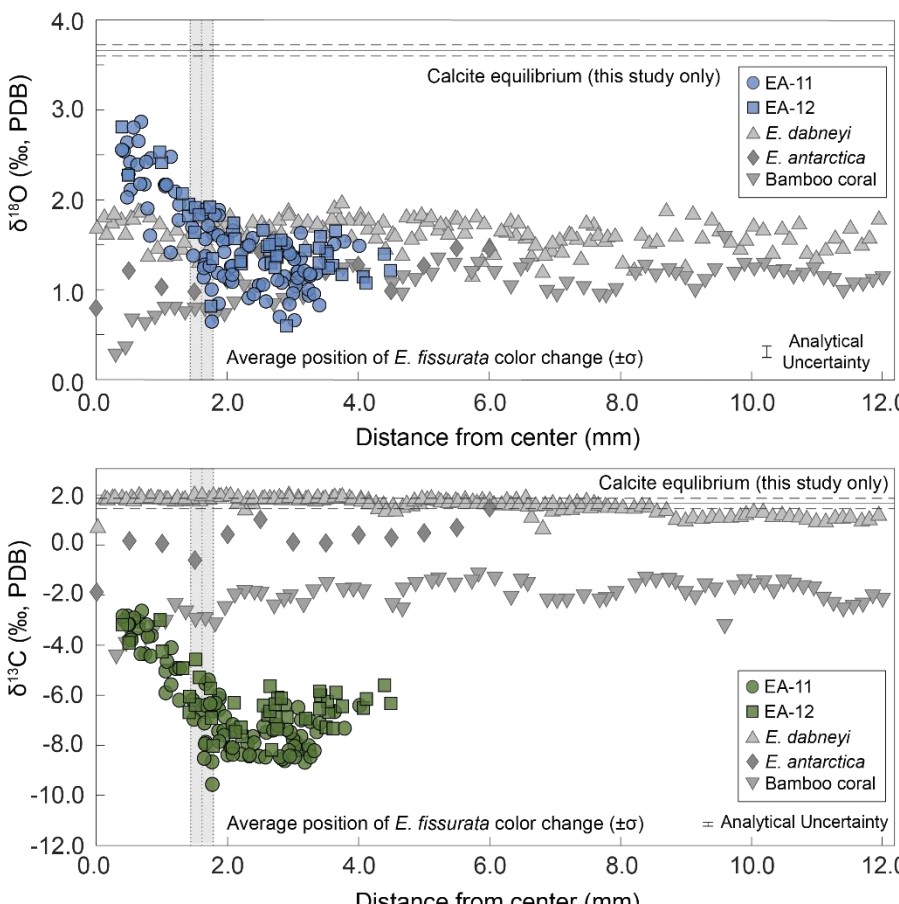

**Figure 7:** Oxygen (top) and carbon (bottom) stable isotope ratios by distance from the center of the coral. The values from this study are plotted by slice from each specimen as noted by the colored circles (EA-11) and squares (EA-12). The horizontal lines denote calculated calcite equilibrium values for this study with corresponding uncertainty in dashed lines (±1σ). Vertical gray bars denote the average radius of the white coral center for *E. fissurata*, the width of the bar denoting ±1σ. Also included are additional coral data sampled in a similar transect style. *E. dabneyi* is from Wisshak et al. (2009), *E. antarctica* is from Samperiz et al. (2020), and the bamboo coral data are from Hill et al. (2011). Regarding δ¹⁸O, the *E. antarctica* and bamboo coral seem to have a small decrease in the last few datapoints toward the center. Regarding δ¹³C, the same is true of each of the corals from other studies. This contrasts the increase we observe in our corals.

isotopic compositions. This supports that vital effects can have a similar influence on corals of different classes, and across different latitudinal and depth ranges. Unlike the scleractinian corals analyzed by Adkins et al. (2003), *E. fissurata* does not exhibit a "break" in the slope of the linear regression (Fig. 4). This is supported by the findings of Stewart et al. (2022) wherein

boron isotopes revealed that stylasterids do not upregulate the pH of their internal calcifying fluids to enhance calcification, thus altering the linear relationship between isotope ratios. An unusual isotopic trend from the *E. fissurata* corals is the innermost values of each sample disc approaching equilibrium (Figs. 3, 5, and 7). This contrasts the findings of Samperiz et al. (2020) who measured the innermost portion of *E. antarctica* as furthest from isotopic equilibrium (Fig. 7; note that the equilibrium value in this figure is for this study only, however, the *E. antarctica* isotope ratios clearly decrease towards the coral center). Additionally, Hill et al. (2011) sampled bamboo corals across a transect which resulted in $\delta^{18}O$ and $\delta^{13}C$ values further from equilibrium in the center of the coral (Fig. 7). Results from Wisshak et al. (2009) can only partially support these works as they generated $\delta^{13}C$ values from the center *E. dabneyi* that produced a single value from the innermost portion that was much lower than equilibrium (Fig. 7). Stable isotope ratios closer to equilibrium in the center of *E. fissurata* are the first of their kind for stylasterids and we posit that they imply lessening vital effects in this region because of slower calcification rates (McConnaughey, 1989b). Figure 7 also illustrates the wide range in isotopic variability exhibited by *E. fissurata* compared to the aragonitic *Erinna* and calcitic bamboo corals (Wisshak et al., 2009; Hill et al., 2011, respectively). The range in isotope values that trend toward equilibrium in the center of the coral could be a result of a unique calcification strategy that also leads to a lack of visible banded structure within *E. fissurata*.

Rapid calcification rates, and therefore prominent vital effects have been interpreted as the reason that *E. antarctica* growth tips exhibit some of the lowest $\delta^{18}O$ and $\delta^{13}C$ values when compared to main trunk and branch samples (Samperiz et al., 2020). Here, the most negative $\delta^{18}O$ and $\delta^{13}C$ values for EA-11 were measured on the slice closest to the tip, EA-11a, and the most positive $\delta^{18}O$ and $\delta^{13}C$ values were measured on a slice furthest from the tip, EA-11d (Fig. 4). Specimen EA-12 was characterized by $\delta^{18}O$ and $\delta^{13}C$ minima nearest the tip (EA-12a), but the maximum isotope ratios were measured on different slices: the most positive $\delta^{18}O$ was closer to the tip and the most positive $\delta^{13}C$ was further away (Fig. 4 and Table 2). These observations, however, are not a direct comparison to measurements by Samperiz et al. (2020) for two reasons. First, the sampling scheme here was designed to exploit isotopic variability across a coral slice, thus considering calculated "bulk" values from each slice should provide more comparable value. However, our calculated bulk values do not exhibit any significant changes over the lengths of the corals; they all exhibit similar variance along their respective coral and are within error of each other (Fig. 4). This could be due to the second limitation on comparisons: we did not sample the true tips of the corals, but instead were ~1 cm below, possibly preventing us from sampling the most negative $\delta^{18}O$ and $\delta^{13}C$, so we cannot truly compare growth tips. We are also uncertain of the lower extent of EA-11 and EA-12, as they were dredge collected, and could be missing a portion of the main trunk, possibly preventing us from sampling the most positive $\delta^{18}O$ and $\delta^{13}C$ (Fig. 2).

An additional influence on skeletal $\delta^{18}O$ and $\delta^{13}C$ values of stylasterid corals is carbonate mineralogy. It has been widely known that fractionation of stable oxygen and carbon isotopes occurs between carbonate and seawater, the magnitude of which is different for aragonite and calcite (e.g., O'Niel et al., 1969; Grossman and Ku, 1986; Swart, 1983; Romanek et al., 1992). This has been observed in laboratory settings, as well as organisms that precipitate carbonate shells or skeletons (Romanek et

al., 1992; Lécuyer et al., 2012; Samperiz et al., 2020, respectively). In addition to taxonomical organization, mineralogical content has recently become a common method by which to categorize specimens when analyzing skeletal geochemistry (Stewart et al., 2020; Samperiz et al., 2020; Stewart et al., 2022; Kershaw et al., 2023). The *E. fissurata* here were determined to be mixed mineralogy; mainly calcite with some aragonite, and likely a slightly greater proportion of aragonite in the white center compared to the pink outer region (Fig. S3 in the Supplement; Table 3). The mineralogy data reported in Table 3 are from specimens EA-13 and EA-14 which, although were collected from the same dredges, are not the corals analyzed for stable isotope values, and therefore might be characterized by slightly different calcite and aragonite proportions. With such apparent mineralogical variability among the *Errina* genus (see Fig. 6; Cairns and Macintyre, 1992), it would be prudent to assume some variability among individuals within a species. The sample that was analyzed for both stable isotopes and mineralogy, EA-11d, was scanned for XRD over the entire slice as a bulk measurement, as well as two outer pink samples and one white center sample (see Fig. 3 for EA-11d image; Fig. S3 in the Supplement). The bulk sample would be most comparable to the bulk mineralogy data reported in the literature, but the subsamples of coral color bands are the first of their kind. The peak intensities from the coral color bands show some differences between the pink and white, however, the scan of the "right side" pink band may have incorporated some of the white center as the coral did not produce completely straight demarcations (see EA-11d in Figs. 2 and 3). This work presents the first skeletal stable isotope values collected from mixed mineralogy stylasterids, and further, the first stylasterid mineralogical data obtained by methods other than bulk sample analysis. Despite mixed mineralogy, the *E. fissurata* $\delta^{18}O$ and $\delta^{13}C$ values align well in isotope space with other calcitic stylasterids; they are further offset from isotope equilibrium than their aragonitic counterparts along the $\delta^{13}C$ axis, but within range of the aragonitic $\delta^{18}O$ values (Fig. 6).

## 4.2 Surveying maps of skeletal stable isotope composition: alternative mechanisms for high central $\delta^{18}O$ and $\delta^{13}C$ values

### 4.2.1 Organic contribution to outer skeletal portion

Stylasterid corals have been observed hosting other species including boring cyanobacteria, sponges, and gastropods, all of which could affect the isotopic composition of the coral skeleton (Puce et al., 2009; Braga-Henriques et al., 2010; Pica et al., 2015). The presence of such organisms could contribute metabolic carbon and/or oxygen to the pool from which the corals calcify, similar to the metabolic effects of shallow-water coral symbionts (e.g., McConnaughey, 1989a). Depending on the relationship of the epiphyte, and if it is living near an active calcification site, isotopically light metabolic products could be incorporated into the coral carbonate, driving the outer portions of the coral to lower $\delta^{18}O$ and $\delta^{13}C$. There have been direct observations of barnacles living on *E. fissurata* (Pica et al., 2015). For specimen EA-11, sections with any obvious external growth or signs of hosting were avoided and only clean slices were selected for sampling. For EA-12, however, this coral was believed to be dead upon collection, and covered in possible organic growth/encrustation over time (Fig. 2). We posit that this material was not present at the time of coral calcification and therefore would not have altered the $\delta^{18}O$ and $\delta^{13}C$ values of the carbonate. The similarity in isotope trends between both specimens supports this hypothesis.

### 4.2.2 Diagenetic influence on stable isotopic composition

Coral diagenesis can be described as a post-calcification alteration of carbonate geochemistry. Such processes could directly change the skeletal $\delta^{18}O$ and $\delta^{13}C$ values by removing or replacing the carbonate. Identifiable diagenetic features for stylasterid corals have been described by Black and Andrus (2009), including encrustation of foreign materials on the outer surface of the coral skeleton, microspar fabric and microbioclastic debris, abrasions, fragmentation, pitting, and bioerosion. These features are identifiable via scanning electron microscopy (SEM), but the authors found no impact on the $\delta^{18}O$ and $\delta^{13}C$ after analyzing

fossil and modern corals (Black and Andrus, 2009). Work by Wisshak et al. (2009) has identified additional diagenetic impacts on stylasterid corals via micro-scale dissolution and re-precipitation. We were able to analyze *E. Fissurata* live-collected specimens EA-22, EA-23, and EA-24 for physical signs of diagenesis. Although we did not image specimens EA-11 or EA-12, the others were collected from the same dredges. We imaged slices of coral branches similar to the stable isotope sampling scheme and did not see any signs of the diagenetic effects described, except for possible secondary growth or recrystallization

(Figs. S4 and S5 in the Supplement). Although we have identified evidence of secondary crystal growth, the scale is on the order of ~100 µm. That size is 5 times smaller than the size of the drill bit used here, and therefore unlikely to have an impact on the $\delta^{18}O$ and $\delta^{13}C$ variability observed.

### 4.2.3 Calcite versus aragonite mineralogy

With the XRD data presented here, there are multiple lines of
evidence for mixed calcite and aragonite *E. fissurata* skeletons. This is important to consider because the mineral phase of carbonate can influence isotopic fractionation, as described above. Comparing $\delta^{18}O$ and $\delta^{13}C$ values from eight specimens of *Cheiloporidion pulvinatum* (five purely aragonite and three purely
calcite) collected from the same location resulted in an average difference of ~ +1.4 ‰ $\delta^{18}O$ and ~ +5.3 ‰ $\delta^{13}C$, the higher values

| Coral | Sample Location | Aragonite | Calcite |
|---|---|---|---|
| EA-13 (Live) | Center white | 23% | 77% |
| | External pink | 20% | 80% |
| EA-14 (Dead) | Center white | 30% | 70% |
| | External pink | 25% | 75% |

**Table 3:** Results for XRD mineralogical analysis of *E. fissurata*.

being aragonite (Samperiz et al., 2020). Despite the quantitative XRD analyses being performed on corals not used for stable isotope measurements, we cannot eliminate mineralogy as a potential driver for the change in isotope composition toward the center of the corals. Two *E. fissurata* corals were sampled here (one modern and one dead) from both the center white, and
outer pink regions. Each coral sample produced similar results of ~75 % calcite, and ~25 % aragonite (Table 3). There is, however, a small change in the percentage of aragonite from the outer portions of both corals to the inner portions wherein they increase by magnitudes of 3 % (EA-13) and 5 % (EA-14) towards the center. These are not large changes and may not produce the magnitude of isotopic differences observed between pure aragonite and calcite by Samperiz et al. (2020). However, these are the first $\delta^{18}O$ and $\delta^{13}C$ records from stylasterids of mixed mineralogy so we must consider a mineralogical driver and
pursue it further as an avenue to understand *E. fissurata* biocalcification and use as a paleoceanographic archive.

### 4.3 Hypothesized large-scale calcification model for *E. fissurata*

To further investigate the possibility of vital effects leading to specimen-scale differences in $\delta^{18}O$ and $\delta^{13}C$ values, we posit that *E. fissurata* calcifies in a way that magnifies the vital effects in the outer region compared to the center. In order to grow dendritically, the most intuitive growth model for corals such as our stylasterids holds that the center of each branch extends axially faster than each branch thickens radially. Thus, the kinetic effects would be most apparent in the center of each coral slice and lessen towards the outer edges, resulting in $\delta^{18}O$ and $\delta^{13}C$ values that would therefore approach equilibrium toward the edges of each isotope map. Many marine calcifying organisms (e.g., mollusks and corals) follow the Von Bertalanaffy growth model wherein their calcification rate ontogenetically decreases, and we posited that this model also applied to the horizontal extension of stylasterids as they prioritize vertical growth (Ralph and Maxwell, 1977; Emiliani et al., 1978; Berkman, 1990; Philipp et al., 2005; Román-González et al., 2017). Such a coral growth model is supported by observations from Fallon et al. (2014) wherein a cold water scleractinian coral demonstrated growth similar to stacking cones with a fast growing tip. This structure is also supported by isotopic records of other stylasterids and bamboo corals which were all characterized by low isotopic values at the regions of most rapid growth (Wisshak et al., 2009; Samperiz et al., 2020; Hill et al., 2011).

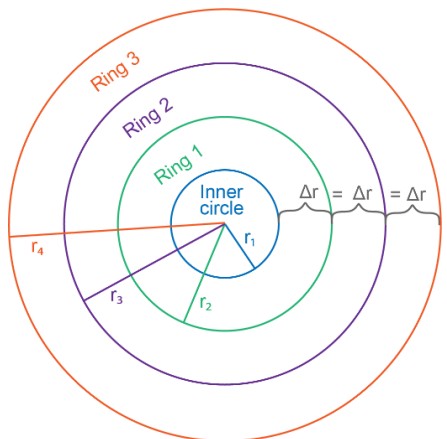

**Figure 8:** Coral slice schematic used for calcification models. In this simplified model, growth begins with the inner circle of radius $r_1$. Each growth increment is noted by a new ring, Ring n, with a radius, $r_{n+1}$. In this case, the change in radius between each ring, $\Delta r$, is equal representing the constant radial extension model. The $\Delta r$ becomes larger or smaller with each ring for the increasing and degreasing growth scenarios, respectively.

To better understand our resulting trends, we created models that reflect simplified calcification scenarios of constant, decreasing, and increasing radial extension with time. The model calculations were based on an idealized coral slice of n rings with radii, $r_{n+1}$, an initial growth inner circle of radius, $r_1$, and the extension rate represented by the change in radius, $\Delta r$, between each ring (Fig. 8). We set each model to run for 100 years and examined the rate of change of coral area as a proxy for calcification rate in each scenario. Whereas few studies have determined vertical growth rates of deep-sea stylasterid corals (Stratford et al., 2001; Chong and Stratford, 2002; Miller et al., 2004; Wisshak et al., 2009; King et al., 2018), their radial extension rates remain largely unknown. Therefore, we employed a range of radial growth rates reported for live-collected bamboo corals living at a depth of 1000–1700 m (Thresher et al., 2011). The bamboo corals were reported to grow from 0.01–0.12 mm $y^{-1}$; therefore, our constant radial extension scenario was characterized by the median 0.065 mm $y^{-1}$ (Fig. 9a–c). The decreasing radial extension scenario was initially set to the maximum extension rate of 0.12 mm $y^{-1}$ that decreased linearly by 0.0011 mm $y^{-2}$ until the minimum growth rate of 0.01 mm $y^{-1}$ was reached at 100 years (Fig. 9d–f). For the increasing radial

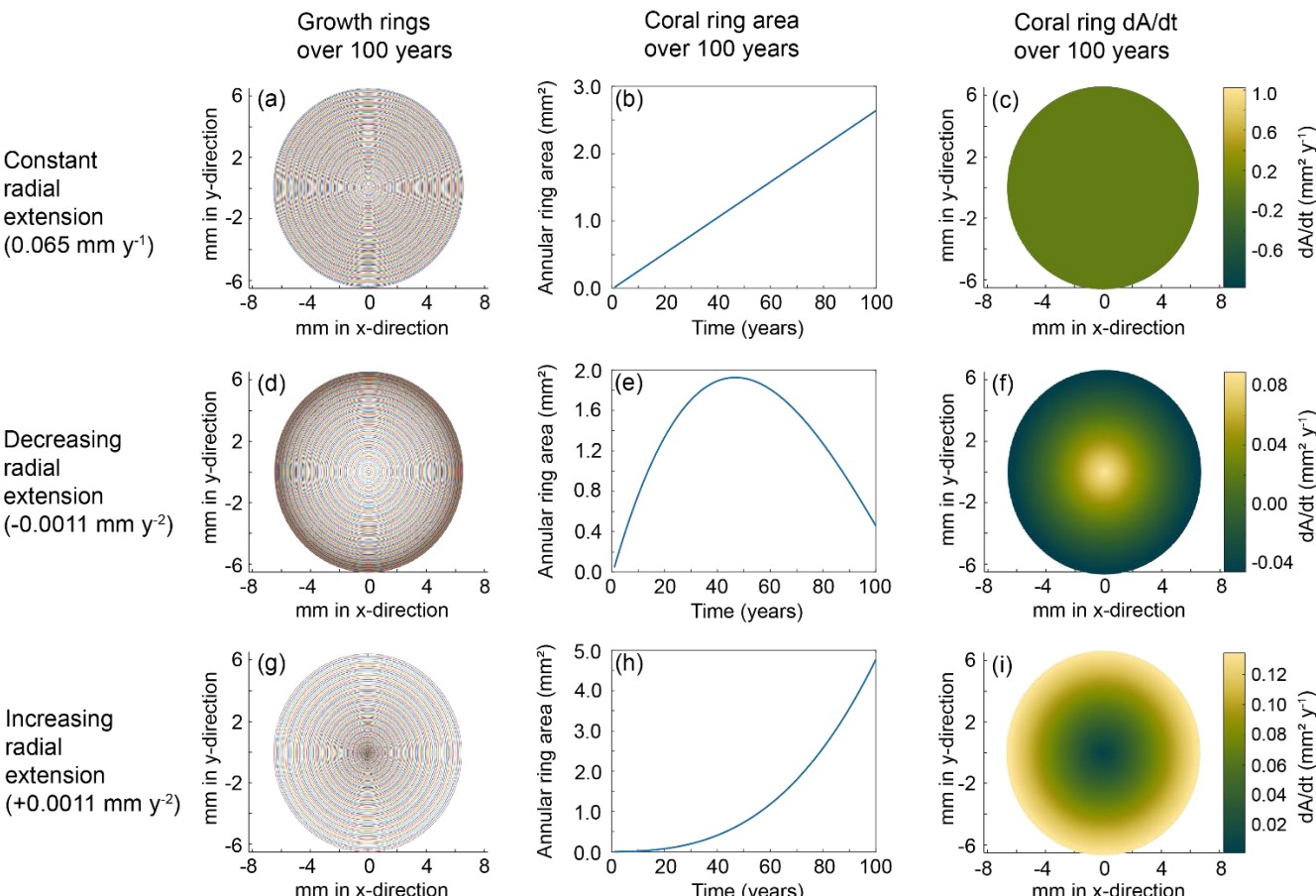

**Figure 9:** Coral calcification models for each scenario. The top row (a–c) depicts results from the constant radial extension model, the middle row (d–f) is the decreasing radial extension, and the bottom row (g–i) is the increasing radial extension model. The left column depicts the growth rings for each model after a run of 100 years. The middle column shows the change in area with each year of calcification. The right column shows the rate of change in coral area for each model. The corresponding color bars represent the magnitude of change.

extension scenario, the opposite was set, an initial radial extension rate of the minimum 0.01 mm y$^{-1}$ and a linear increase at a rate of 0.0011 mm y$^{-2}$ until 0.12 mm y$^{-1}$ at 100 years (Fig. 9g–i).

We observe stronger vital effects toward the outer edges of the *E. fissurata* coral slices and interpret this as increased calcification rate along the outer edges. To test if this growth scenario likely represents our stable isotope maps, we defined changes in calcification rate as the two-dimensional changes in coral area with time. We then compared the resulting models to the changes in stable isotope values across each coral slice (assuming no change in the ambient seawater values during the time of precipitation). In the case of the constant radial extension, the growth rings are evenly spaced, leading to a steady

increase of horizontal area between the rings with time (Fig. 9a and b). Constant radial extension refers to constant change in area with time (Fig. 9c). Such a growth model would result in stable isotope values to either be constant across the surface of

the coral or record any changes that reflect seawater variability. Decreasing radial extension with time relates to decreasing area with time (Fig. 9d–f) and would be the most likely calcification pattern as the available literature supports slowing growth with time among marine carbonates (Ralph and Maxwell, 1977; Emiliani et al., 1978; Berkman, 1990; Philipp et al., 2005; Román-González et al., 2017). With decreasing radial extension, the growth bands become closer with time, causing the area to increase then decrease over the modeled 100-year span (Fig. 9d and e). Therefore, the rate of change of area decreases toward the outer edges of the coral surface (Fig. 9f) and the center of the coral would exhibit isotope values furthest from equilibrium. That is the opposite of what we observed with *E. fissurata*. The increasing radial extension scenario led to growth bands increasing distance from each other, resulting in an exponential increase in area (Fig. 9g and h). The resulting calcification scheme (i.e., rate of change in area) would lead to stronger vital effects toward the outer edges of the coral surface and isotopic values closer to equilibrium in the center (Fig. 9i) as observed with *E. fissurata*.

If driven by calcification scheme, the unique pattern presented here of the highest isotopic values in the center of the coral with lower values toward the mid and outer edge supports a model of rapid calcification toward the outer region and slower calcification in the center. Such a model, however, is not supported by previously mentioned growth habits of marine carbonates and would not be expected to result in dendritic corals. To reconcile these contradictions, we explore calcification strategies that invoke uneven, or two-step biomineralization to account for the spatial distribution of vital effects observed in our coral isotope maps.

A growth model of shallow water staghorn coral (*Acropora cervicornis*) over daily to yearly timescales reconciles our data (Gladfelter 1982, 1983, 1984). The Gladfelter model includes two main phases: rapid extension by randomly oriented crystals that develop the main skeletal framework, followed by infilling and strengthening of the skeleton along the entire growth axis (Gladfelter, 1982). Gladfelter (1983) observed that the initial framework growth was largely around the mid and outer regions of the corallite and the infilling occurred later in the center. We posit that the basic premise of this model (rapid initial growth followed by slower calcification) describes the coral growth presented here. The lower isotopic values toward the outer edge signify enhanced kinetic fractionation from the rapid growth. The center region of the coral calcifies at a slower rate; thus, kinetic fractionation is reduced resulting in higher isotopic values closer to equilibrium. We acknowledge that the Gladfelter growth model employs fusiform crystals setting a foundation upon which bundles of aragonite crystals grow, and we are not able to determine that scale of calcification here. We instead posit that because of our consistent spatial isotopic discrepancies, and our contrast to previous stylasterid works, there must be a heterogeneous growth structure for *E. fissurata* not previously described for stylasterids, and the growth described by Gladfelter (1983) does this.

A final consideration for growth mechanisms driving the $\delta^{18}$O and $\delta^{13}$C trends in *E. fissurata* is that of the internal structures described for stylasterids. The stylasterid *Errina (Errina) labiata* was originally described by Moseley (1878). Internal, decalcified structures of *Errina* were described in this work including the coenosarcal mesh, which is a network of small canals just below the coral surface that increased in size and distance from each other deeper into the coral (Moseley, 1878). Wisshak

et al. (2009) also observed these structures in their analysis of *E. dabneyi* microstructures. Vacuum-resin-embedded and decalcified coral branches were observed to have very dense meshwork of canals near the outer surface of the corals which became wider, deeper inside (Wisshak et al., 2009). Puce et al. (2012) employed volume rendering methods using X-ray computed microtomography to identify microstructures of *Distichopora* specimens. This work agreed with the canal networks that are small near the surface of the coral and larger inside (Puce et al., 2012). Combined, these works support an uneven calcification mechanism for these few stylasterids that could also apply to *E. fissurata*. The small, densely packed meshwork near the outer edges of the coral surface would have much more surface area than the larger, less dense canals in the coral interior. Therefore, the outer regions would require faster calcification rates to keep up growth of the entire coral colony. This uneven rate of calcification would impose vital effects in the outer regions of *E. fissurata* if a similar growth pattern were occurring, leaving the center region closest to isotopic equilibrium.

## 4.4 Considerations for paleoceanographic reconstructions

The definitive trend of increasing isotopic values toward the inner, white section of the coral supports that this is the ideal region of the coral from which to sample *E. fissurata* for paleotemperature reconstructions. Because the higher values are located nearest the white section of the coral and not the geometric center, we converted the images of each slice to grayscale (Fig. 10). This allowed us to find the whitest pixels and quantitatively determine the most ideal location for sampling. With the exception of the light outer edges, the sample holes that were at least 75 % surrounded by the whitest pixels were selected for temperature reconstructions (Fig. 10). These $\delta^{18}O$ values were averaged for each coral slice and, in the absence of a mixed mineralogy $\delta^{18}O$ temperature calibration for stylasterids, we use the temperature calibration equation from Samperiz et al. (2020). This equation was established for 100% aragonite stylasterid corals and it is unclear if this is appropriate for our mixed calcite and aragonite *E. fissurata*. Additionally, the $\delta^{18}O$ average for each entire slice was calculated to determine a "bulk" value that would represent a common sampling method of drilling across the growth axis. The bulk $\delta^{18}O$ values were also

calibrated to temperature and the results are illustrated in Fig. 10 (Table S3 in the

Supplement). Note that for slice EA-11d that consists of a vertical face rather than

horizontal, $\delta^{18}O$ values were only combined for the same distance from the coral tip.

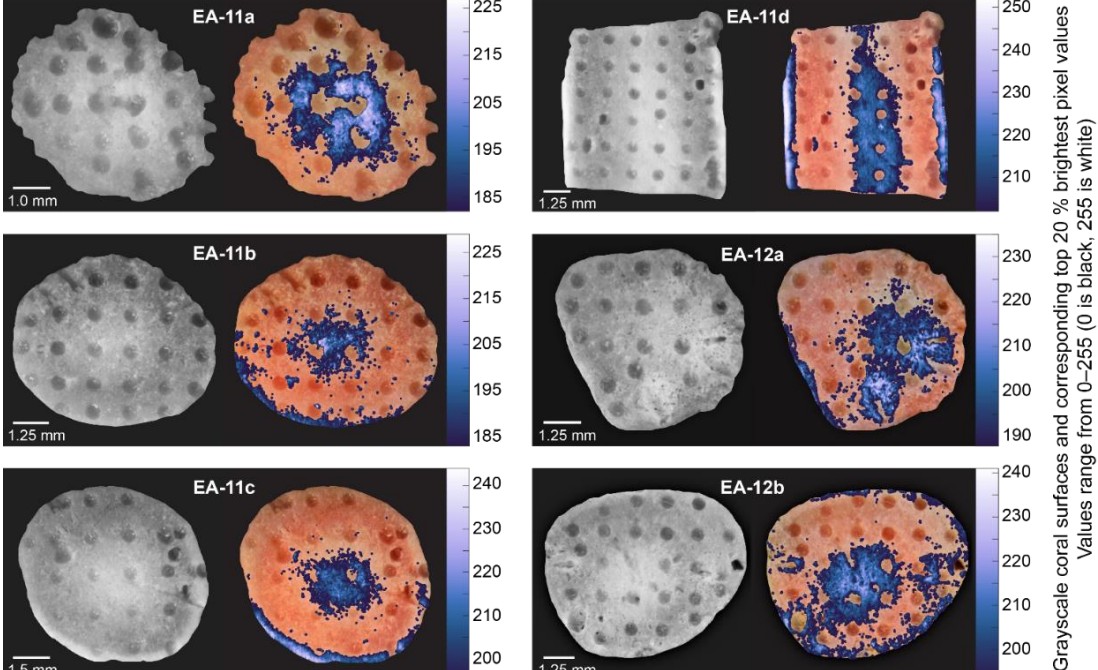

**Figure 10:** Compilation of grayscale images and corresponding brightest pixels (highest 20 %) over each coral slice. The colors correspond to the grayscale pixel value (higher on the scale bar denotes a whiter pixel). The sample holes surrounded by at least ~75 % of bright pixels were used to calculate representative $\delta^{18}O$ values for each slice for temperature calibration (Table S3 in the Supplement).

Although both the "central" and "bulk" temperature records are higher than current ocean temperatures of a similar depth, the center values are significantly different from bulk

measurements and are closer to the true environmental signal (Fig. 11). We compared the center and bulk temperature records using the Student's t-Test to determine whether the difference between the means is significant or occurred by chance (from variability). We used a paired t-Test to compare center and bulk temperatures from the same distance from the tip where available. With a p-value of 6.61 x 10$^{-6}$, the difference between center and bulk temperatures is very unlikely to have occurred by chance (Table S4 in the Supplement). This analysis, however, assumes no environmental change over the lifespan of the

corals. To account for possible seawater changes and more accurately reflect the samples that would be targeted for temperature reconstructions, we also tested individual temperatures across each slice (rather than the averages used in Fig. 11). The unpaired t-Test required at least two temperature values that were sampled from the white center and the pink outer regions for at least one degree of freedom. This was met by four coral slices, EA-11a, EA-11b, EA-11d (63.55 mm from the tip), and EA-12b; each test producing p-values less than 0.005 (Table S4 in the Supplement). These analyses show that we can conceive of no

comparison within coral slices that show the same values between the center and bulk temperatures.

Figure 11 shows that detailed analysis of isotopic distributions through *E. fissurata* skeletons does not quite achieve samples representative of isotope equilibrium; but closer approach is possible. With the aid of computerized tomography (CT) scanning methods, perhaps density banding skeletal structures could be visualized, enabling even finer-scale samples . We posit that our prescribed sampling scheme for these corals, paired with CT-guided sampling, can yield robust paleotemperature records. Sampled as we have, with even distribution of samples over the surface of each disc as a goal, our reconstructed temperatures are higher than a likely environmental signal (Fig. 11). However, we have demonstrated that the center white region of these corals is less affected by vital effects and less likely altered after initial deposition. Therefore, we recommend sampling of the white center using more spatially precise micro-milling methods to target even smaller regions, thus minimizing the impact of vital effects. We also note that in calculating uncertainty associated with temperature calibrations, our center temperature errors are likely underestimated as they rely largely on analytical uncertainty. However, our calculation of temperatures from bulk data includes high spatial standard deviations; more precise sampling would yield more precise temperature estimates. Thus, a targeted milling approach would likely reflect a more accurate uncertainty estimate on the temperature as well. Additionally, CT scanning that reveals coral growth structures, such as Puce et al. (2011), would allow an even closer approach to accurate paleoceanographic reconstructions. Such combined methods would allow for extremely targeted sampling considering both the growth structure and biomineralization methods to inform sampling efforts to allow closer approach to equilibrium and better accuracy of paleotemperature reconstructions using *Errina* corals.

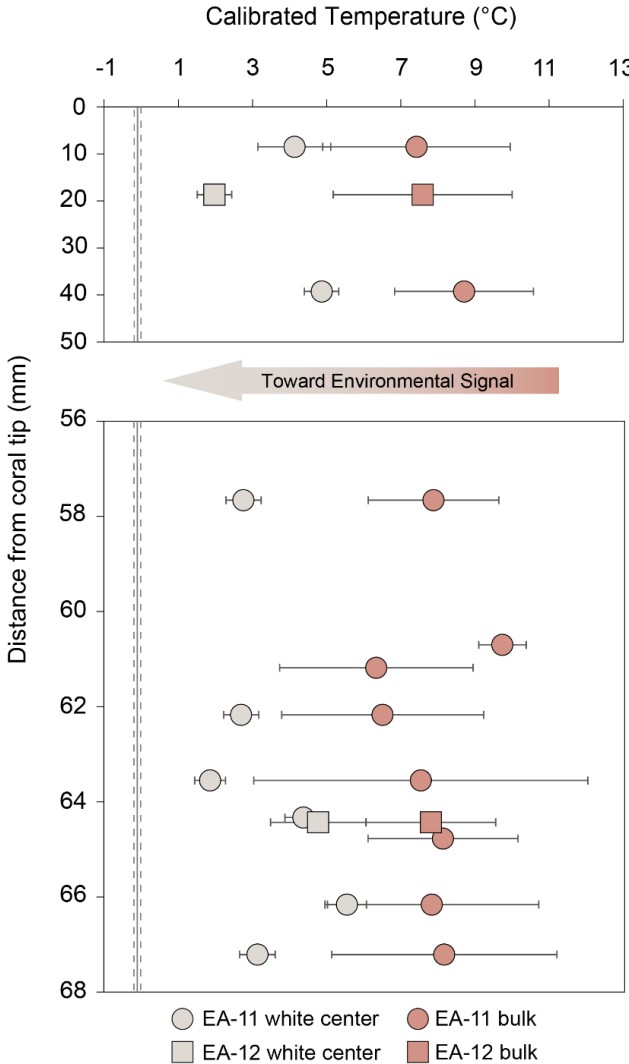

**Figure 11:** Temperature calibration using the equation from Samperiz et al. (2020) for aragonite stylasterid corals. Light colored points denote temperatures using the $\delta^{18}O$ values from the white centers of each slice, whereas the darker, peach color denote a "bulk" measurement. The bulk is calculated using the average $\delta^{18}O$ of the entire slice, similar to a value that would be measured from a sample collected via drilling into the side of a coral specimen. Error bars are based on standard deviation of three or more averaged samples. The error bars for less than three samples are based on the analytical error propagation (Table S3 in the Supplement).

## 5 Conclusions

The results presented here contribute to a new understanding of the highly diverse family of stylasterid corals. In the case of *E. fissurata*, a mixed mineralogy taxon, stable isotopic records provide geochemical details over a unique sampling resolution. We have demonstrated that *E. fissurata* skeletons exhibit significant and variable vital effects that obscure the environmental signal of this potential paleoceanographic archive. Lacking any visible banding structure, the specimens in this study benefitted from a gridded sampling scheme to determine the location of minimum influence from vital effects. Our skeletal $\delta^{18}O$ and $\delta^{13}C$ values exhibit an increasing trend toward seawater equilibrium near the center of the coral. This result contradicts growth structures hypothesized in the current body of literature based on observed stable isotopic trends that have only focused on microsampling purely calcitic or purely aragonitic specimens (Wisshak et al., 2009; Samperiz et al., 2020; Hill et al., 2011). We posit that the observed trend in $\delta^{18}O$ and $\delta^{13}C$ values could be a result of changing mineralogical composition over the *E. fissurata* skeleton or a heterogeneous growth/calcification structure (or some combination of both). Possible growth strategies include (i) a two-stage growth model with a "hasty" initial lattice framework construction, followed by slower infilling to strengthen, and support the coral colony (Gladfelter, 1982, 1983) or (ii) rapid calcification around the outer edges of the coral stem to construct the dense canal meshwork that is more sparse towards the center (Moseley, 1878; Wisshak et al., 2009; Puce et al., 2012). We tested our interpretations that the $\delta^{18}O$ values from the center white region of the corals were closest to equilibrium by calibrating them to seawater temperature using the paleotemperature equation developed by Samperiz et al. (2020) for aragonite stylasterids. These $\delta^{18}O$ values produced temperatures significantly different than a bulk approach and were closest to nearby ocean temperatures. Thus, we recommend sampling this taxon along the center, white region where the carbonate geochemical record is closest to seawater equilibrium and an environmental isotopic signal. Further, we suggest initial CT scanning of corals to determine any hidden growth features, and micro milling sampling techniques to target the location of the most accurate record. Combined with continued efforts to develop mixed mineralogy stylasterid temperature calibrations, we posit that accurate paleotemperature records can be reconstructed from deep-sea *E. fissurata* corals.

## Data availability

All data are available in the supplement.

## Author contributions

TMK and BER contributed to the conceptualization of the work. TMK developed sampling methods under the supervision of BER. NPJ provided expertise on mineralogical analyses. All authors contributed to formal analysis. TMK wrote the manuscript draft, with significant input from BER and NPJ. All authors approved the final version of the manuscript.

## Competing interests

The authors declare that they have no conflict of interest.

**Acknowledgements**

We thank the captain, crew, and scientific staff of the RV *Nathaniel B. Palmer* during the US Antarctic Program expedition NBP07-01 to the Ross Sea, Antarctica for their efforts in coral collection. We also thank Dr. Ernst Peebles for the use of laboratory space during coral sampling, and Drs. Jennifer Granneman and David Jones for guidance on both slicing the corals and initial setup of the MircoMill. We also thank Dr. Lukasz Wojtas for providing XRD analysis at the University of South

Florida and guidance during sample preparations. During this project, TMK was funded through the Genshaft Family Dissertation Fellowship awarded by the University of South Florida as well as three endowed fellowships from the College of Marine Science: Carl Riggs Endowed Fellowship, George Lorton Fellowship in Marine Science, and the Abby Sallenger Memorial Award.

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
