# Peer review of "Deep-sea stylasterid $\delta^{18}O$ and $\delta^{13}C$ maps inform sampling scheme for paleotemperature reconstructions"

_Biogeosciences, 2022_

## Author Comment (AC1)

*We thank the anonymous reviewer for their constructive comments and thorough review of this manuscript. Below are the reviewer's comments and author responses in blue italics.*

The authors explore the stable oxygen and carbon isotope heterogeneity in an extremely under-studied cold-water coral taxon – stylasterids. They present high spatial resolution δ18O and δ13C results from individual corals that show highest values, closest to equilibrium seawater values, in the center of the two corals examined. This heterogeneity implies that potentially large differences in reconstructed seawater temperature could result from sampling different parts of a single coral specimen. The motivation for the paper is sound; it is clearly of great importance to understand the isotopic chemistry of coral skeletons so that they can be accurately used as temperature proxies, and we can understand their biocalcification mechanisms.

The paper uses two examples of the genus Errina, one dead and one alive. The genus Errina has been studied in prior publications e.g. Samperiz et al. (Rosenheim was a co-author on this study), and Wisshak et al., The main findings in the paper directly contradict these earlier studies, so it is particularly important that all options are explored to explain the new observations. For example, it is unclear whether the effects seen in this study are a feature local to this site, or to Errina fissurata. Given the contradictory results between Errina specimens, then it is important to avoid generalized statements about sampling strategy for temperature proxy work (e.g. centers only). Indeed, the current advice to use the white material from the center will not hold in samples which do not exhibit this coloration. The authors hint that this may be a mixed mineralogy species  - mineralogy is likely a key factor for explaining the different findings between studies and needs to be addressed in more detail in the discussion.

*The reviewer brings up a good point here in that we did not explore every option that could affect the stable isotopic records like diagenetic or mineralogic effects. We are editing our discussion to include mineralogical data collected on different coral specimens from the same sample collection. In doing so we are including an additional author, Dr. Noel P. James (Queen's University), who conducted these analyses. These samples were analyzed in parallel with our work, but not included in the initial manuscript because they were not from the same exact specimens. However, they are useful to address the reviewer's comments.*

*In terms of the generalized statement about sampling the center white portion of these corals, we understand that not every coral will have the same coloration. The reviewer mentioning this has demonstrated that we need to clarify that we only recommend this sampling scheme for the corals in this study. We will thoroughly edit our manuscript to reflect this. What is important for others to investigate, however, is whether there are similar isotope patterns in other corals.*

The paper omits to reference or draw on a number of stylasterid publications – including those by the authors, that give some sense of the age (e.g. Millar et al 2014, King et al 2018, Wisshak et al 2009) needed for the modelling, and information on biomineralization and diagenesis (e.g. Black and Angus 2012, Stewart et al 2021)

*These publications were not included in our growth model because we were looking for rates of radial extension for individual branches, not overall vertical/axial extension as described by these works. These publications, and others, note varying vertical growth rates from ~0.2 mm/year to 7 mm/year (King et al., 2018; Miller et al., 2004; Chong and Stratford, 2002; Stratford et al., 2001; Wisshak et al., 2009), therefore, we looked for any other records of deep-sea branching corals for realistic rates of radial extension. However, another reviewer had similar comments and suggested that we incorporate an estimate of radial growth using the distance between radiocarbon dates and the radius of the branches from King et al. (2018). Therefore, we will include this calculation.*

The recommendation is that this manuscript should be published but that the discussion and conclusions should to be revised. Below are some points to be considered.

**Abstract:**

Remove statement that the study identifies the optimal location for sampling for paloeceanographic studies.
*We will edit this statement to clarify that we mean for the corals in this study.*

Consider removing the final line, as it is not clear that the paper will motivate advanced visualization studies.
*As the final line is written, we state that this work will inform the advanced visualization techniques (i.e., inform where to apply computed tomography (CT) scans), not motivate new studies. We have reworded it for clarity.*

**Introduction**

Line 32: suggest "...fidelity of elemental and isotopic records archived..."
*Done*

Line 35: Simplify sentence. "...decades to millennia, which is especially useful to reconstruct ocean circulation patterns during the time that the coral was alive."
*Done*

Line 56: Suggest delete "...and constructed the most comprehensive model for biomineralization to date." Others may disagree that this is the most comprehensive model, and updates have been published in the intervening years.
*Done*

Line 58: "have a strong linear relationship".
*Done*

Line 59: technically corals are trending "away" from equilibrium where they would be if it weren't for biological processes.
*We will clarify this sentence.*

Line 70: this sentence needs to be revised to be clear that the sampling strategy for the stylasterid is similar to bamboo corals, not that the corals are similar.
*We agree, done.*

Line 74: These statements about stylasterids should be reconsidered. These stable isotope values in these corals generally fall closer to seawater equilibrium than the well-studied Scleractinia suggesting that these heterogeneities in stylasterids are relatively smaller (Samperiz et al., 2020; Stewart et al., 2020 EPSL). These findings should act as strong motivation for this study however, as understanding what is causing these small deviations from equilibrium will yet further refine what are perhaps the most promising coral temperature proxies – stylasterid d18O and Li/Mg. The authors should also consider refer to Stewart et al., 2021 Sci Rep. which highlights the contrasting calcification behavior of stylasterids compared to Scleractinia.
*After reading this reviewer's comments, we see how the end of this paragraph concentrates more on the complications of stylasterid corals, and we believe the comments broaden the potential impact of this paper. We will edit this section to include considerations for scleractinian data which demonstrate the potential for these stylasterids. We will include the publications the reviewer suggested.*

**Methods**

Further information is needed on the sampling sites. What are the local d18O and d13C values in seawater? Are there strong gradients? I have been back through a related publication from King and Rosenheim to try and ascertain the connection between sites. I think that some of the samples are from the same cruise or even dredge, but maybe not the same samples. Why is this? Did you look at the samples from the prior publication? I was surprised not to see the age information derived from that paper not being used in this publication, and not included in the reference list. It surely needs to be cross –referenced and compared. One of the main conclusions of that paper was that changes in the water masses could be identified – can we see any of those changes in these two new corals?

*Regarding the local seawater stable isotopic ratios, we don't have data directly from the coral sampling site. Instead, we've used nearby stations for δ¹³C and δ¹⁸O seawater values and the average for the depth range are listed in the supplement. We will edit the supplement to include each discrete depth to provide the reader with additional context. We thank the reviewer for their diligence as these corals were collected from the same cruise and Ross Sea dredges as the previous publication by King et al., 2018, they are just different specimens. The complication lies in that we are not certain from which dredge the corals were collected (D05, D06, D07, D08, or D09). This is because we performed the geochemical analyses after receiving the corals from a now deceased collaborator and we have not been able to locate sample collection information recorded during that cruise. We will clarify the methods.*

*We did not incorporate any of the age information from the previous radiocarbon work for these corals because the radiocarbon was not an independent chronometer. In the previous publication, we demonstrate that the radiocarbon record is influenced by an older water mass impinging on the corals, i.e., the reservoir age changed. The age of the corals that we calculated was only applicable if we disregard that change. If we compare the length of these coral specimens to those in our previous publication, they are similar, and the corals here may span a similar range of time, but we don't have a direct age control. Further, the temperature calibrations we present in Figure 9 shows little evidence for the incursions of Circumpolar Deep Water (which would be a warming signal) during the time the carbonate analyzed was deposited. Therefore, this work neither supports nor contradict the earlier work because it may be too old or too young to have recorded a water mass change.*

A new panel or figure needs to be added showing the water column properties for the two corals, so that it is easy to identify the potential for changes in temperature and d13C over the lifetime of these corals.

*As mentioned above, the seawater data used here is from a station nearby and not exactly where the corals were sampled. We will add the isotopic data through the entire water column to the supplemental material.*

How old was the dead coral? A radiocarbon date would be helpful here.

*An age constraint would be a nice addition to this work, but we don't think it's necessary to explain the isotopic trends observed in the corals.*

Line 110: presumably the 8mm section was "d". Please say this here.

*Done*

Line 155. 'This variability is unexpected is we assume that each slice / cross section reflect deposition at the same point in time' This assumption does not seem to be likely, given that the corals probably live for at least 100 years.

*We agree and have changed the wording here.*

**Discussion**

Figure 4 and general discussion. The authors rightly make comparisons to previously available data. These data also need to be shown on the figure. Relating to the earlier points above, it would also be helpful to compare to bamboo coral and scleractian coral data to highlight the differential amplitude of the ranges in the isotopic data.

*We agree that it would be beneficial to include other data for a direct comparison to our records. We will edit this figure to include data from previous publications.*

Line 225: delete space in 0.6 8 ‰
*Done*

Line 231: This is a crucial point that needs to be expanded. Stylasterid mineralogy can indeed have a significant influence on their geochemistry. The manuscript would benefit from outlining this in the introduction, followed by more detailed exploration in the discussion and as a requirement for analyses in the conclusions. The fact that stylasterids can build skeletons from calcite, aragonite or both is strange and further highlights the need to study these corals. The big question, though, is what is the mineralogy of the Errina fissurata specimens used in this study? Cairns and Macintyre [1992] record an Errina fissurata specimen from a similar (maybe the same?) location to that represented here, which is composed of 91% calcite and 9% aragonite (sample 40 in their paper). Samperiz et al. [2020] showed that calcitic stylasterids have generally lower d13C and d18O than aragonitic stylasterids. Therefore, this raises the possibility that variable mineralogy in these specimens (e.g. increasing percentage of aragonite toward the center of each section) could explain the isotopic trends observed. Could changing mineralogy also reconcile the new data with the contrasting trends found by Samperiz et al. and Wisshak et al.? The authors hint at some data surrounding the mineralogy of these specimens, and the manuscript would benefit from more detailed discussion of this theme.

*Yes, we completely agree here. We had previously acquired some mineralogical data, but not from the same specimens that were analyzed for stable isotopes (data herein) or radiocarbon (from King et al., 2018). However, the samples analyzed for mineralogy were from the same cruise and dredge(s). After reading these comments, and similar comments from another reviewer, we feel that comparing data from two specimens is both warranted and useful. We will edit the discussion to include the mineralogical data and discuss how that affects stable isotopes. We can quantify the mineralogical change in percentage of calcite-to-aragonite from the outer portion to the center. As mentioned above, this had led us to also include Dr. Noel P. James (Queen's University) as an additional author for measuring mineralogy and facilitating the incorporation of these data into the manuscript. We can use the small change in percent calcite-to-aragonite by region (~3% more aragonite in the interior compared to outer samples of coral slices) for an interesting thought experiment. If mineralogic changes between the outer and inner sections of the skeleton we the most important factor in explaining our isotope variation across skeleton discs, the isotope composition of the new material would have to be on the order of ~-32.5‰, far in excess of any skeletal variability we observed herein. Such a negative value seems unrealistic to be the only forcing of the higher values towards the centers of each of our coral slices. Therefore, with the caveat that the mineralogical data are from a different specimen, we will discuss that mineralogy is an important consideration, however, not the main forcing of the trends we observe. Only a small amount of additional "new" calcite is needed to be within our observed range of skeletal isotope composition, however the mineralogical data do not support a larger change in mineralogy.*

Line 232: see above, suggest changing "additional work needs to be done to approximate the mineralogy over the sampled discs" to focus on how mineralogy might change across the sampled discs.

*Again, we agree and are including additional mineralogical analyses.*

Line 255. Using a bamboo coral growth rate is not appropriate here. As the authos point out, Stylasterids fall within the Hydrazoa, whereas bamboo corals fall within the Anthazoa. They are very different organisms which calcify very differently to one another. There are some age data for stylasterid corals – including data from the authors of this paper (King et al 2018) and others. These need to be cited and used rather than drawing on bamboo corals. Likewise on Line 245 it is not clear that the data from scleractinia can support the biomineralization models proposed given the large differences between them.

*As mentioned above, to calculate the calcification models we needed rates of radial extension for the coral stems and individual branches, not overall vertical extension as described for stylasterids (e.g., King et al., 2018). Therefore, we chose to rely on the bamboo coral data as they were the only we could find describing the radial extension of deep-sea branching corals. Additionally, the specific value of the growth rate doesn't need to be completely accurate for the purpose of demonstrating the growth patterns we were hypothesizing. We only needed a constraint that was likely in the range of possible values.*

Line 275 – Which data point to a slow down in growth rate for stylasterids, please provide reference.
*The line mentioning "the available literature supports slowing growth with time" is referring to the previous paragraph where we describe that marine calcifying organisms generally slow their growth with time. We will reword this to clarify the meaning and cite appropriate references.*

Line 280: The biocalcification modelling is a nice addition, and explains the results of this study, but currently this section doesn't appear to reconcile the results of this study results with the highly contrasting findings in Samperiz et al., and Wisshak et al., who both find stable isotope values tending towards equilibrium towards the outer edge of Errina sp.samples. More discussion/explanation is needed here. Is this a sampling / species / mineralogical / ontogenetic / location effect? Not all samples have the same white / pink delineation, making the proposed sampling strategy challenging.
*We appreciate this comment, but the modelling was not constructed to reconcile the results here with published results. The models were to represent the trends we expected based on those published works and how we would expect the growth to happen based on the trends we observed here. We are adding discussion about mineralogical analysis, and we will also add discussion about a possible species affect. We know that many stylasterids exhibit visible growth rings, which demonstrates the variability among the taxon. We are aware that not all are stylasterids have the white/pink delineation as well. Therefore, we will be sure to clarify our discussion where we recommend the sampling scheme for this widespread species but may not be suitable universally.*

Line 301: delete extra full stop
*Done*

Line 304: The authors could consider comparing their suggested mode of growth – fast initial growth to form a framework followed by slower growth focused in the centers of branches – with that suggested by Wisshak et al. [2009]. Wisshak et al. show evidence for skeletal reorganisation – including dissolution and re-precipitation - during stylasterid growth, following the initial skeletal precipitation. Although the isotopic trends and interpretations of growth models may differ, overall this study and Wisshak et al. appear to be suggesting similar processes. More discussion of this here could lend more support to the authors ideas. It would also be really interesting to know whether there is any visual support for the suggested slow infilling around the initial framework, starting in the central region of each disc? It's hard to see from the photos in figure 3, but is there anything in terms of pore size, structure or general appearance of the carbonate which supports this theory?

*Yes, we agree and will add discussion incorporating the growth described by Wisshak et al. (2009) and possible diagenetic effects as this work has been essential to stylasterid studies. We don't have any visual data but will be conducting SEM imaging soon to investigate. We will look for any aragonite crystals in different regions of the corals and compare that to previous work as well as our isotopic records. The results will be incorporated in the updated manuscript.*

Line 315: While the results of this study alone imply that the best place to sample thesecorals for temperature proxy work would be the centers, this result is not applicable to all Errina specimens.  Samperiz et al. (including authors from this paper), and Wisshak et al., studies find the exact opposite to be true. Until the cause of this discrepancy is established it is premature to suggest a recommended sampling strategy, especially given different banding and coloration patterns in different specimens.
*As mentioned above, we will be sure to edit the wording so that we specify our sampling recommendations only apply to the species analyzed here.*

Line 365: It is not clear that the new results assuage hesitation surrounding the influence of vital effects. The studies by Samperiz et al., 2020 and Stewart et al., 2020 already did this when they showed that bulk sampling of stylasterids provided highly accurate d18O and Li/Mg temperature proxies compared to the existing scleractinian coral calibrations in the literature. The current study has shown that internal heterogeneity and vital effects in stylasterids are more complicated than previously thought and more study is needed to ascertain if this is because of mineralogy / species / location etc
*This reviewer has made a few comments to this affect, and it demonstrates to us that we need to clarify the language throughout the manuscript. We will be sure to assess every time we mention "stylasterids" as a taxon and be sure that they are not used as an umbrella term when we mean to refer to one species.*

Figure 1: It would be useful to have the coral ID's labelled on the figure rather than the dredges. Also, there are 5 dredge sites and just two corals used in this study which is confusing. It would be very useful to compare the sites in King et al 2018. An additional panel or figure is needed with the water column data and indications of the variations we might expect in d13C relating to different local water masses
*We will adjust the figure to label the coral ID's and not the dredge numbers to reduce confusion. As mentioned above, we are unable to determine the specific dredges from which the corals were recovered. We also mentioned above that we used nearby stations for $\delta^{13}C$ and $\delta^{18}O$ seawater values for our equilibrium calculations. We're not convinced that including a water column profile in the figure will benefit the readers as we won't have the values from the exact study location. We will, however, edit the supplement to include each discrete depth to provide the reader with additional context.*

Figure 3: "largest isotopic values" should be "highest isotopic ratios"
*Done*

Figure 4: this figure should include previously published data.
*We will add additional data to this figure.*

Figure 5: Similar microsampling data by Samperiz et al., and Wisshak et al., should be included on this plot for direct comparison of absolute values between studies (e.g. Fig 7 Samperiz et al., 2020). This will highlight the contrasting isotopic results for the central part of the coral in these studies and the current study. It will also show how low the d13C values are in this study compared to the other measurements of Errina in the literature.
*We will add these data to the figure.*

---

## Author Comment (AC2)

*We thank the anonymous reviewer for their constructive comments and thorough review of this manuscript. Below are the reviewer's comments and author responses in blue italics.*

Stable isotope measurements (d18O and d13C) from deep-sea corals provide valuable paleotemperature reconstructions through the water column at all latitudes (unlike shallow-water corals, geographically constrained). Yet, deep-sea scleractinian and bamboo corals have shown complications derived from "vital effects" that deviate the environmental signal in the isotopic records. A new coral taxon (Stylasteridae) has been considered as an alternative archive, as "vital effects" in these specimens have been reported to be lower than those in other deep-sea coral taxa. This study, however, shows that "vital effects" and their impact on the skeletal isotopic composition of stylasterid corals might not be as straightforward as previous studies have shown.

This study carries out fine stable isotopic (d18O and d13C) mapping on several cross-sections of two specimens of Errina fissurata. Results show that sections closer to the growing tip of the colony present more depleted d18O and d13C than sections further down the branch. Equally, those samples located near the centre of each cross-section (branch) showed values that were closer to equilibrium than those samples from the outer areas of the cross-section. Importantly, these results contradict observations from previous papers. The authors present a growth model where an initial skeletal framework form the main structure of the branch (quick biomineralization) followed by slow mineralization of the inner sections for structural strength and argue that this is the source of isotopic differentiation across all cross-sections. This work points towards the need of a deep understanding of the growth mechanisms of Stylasterid corals (or Errina sp. in particular) in order to obtain more precise paleoreconstructions and introduces an strategy (isotopic mapping) to locate the skeletal area closest to equilibrium, and therefore the areas to sub-sample for the aforementioned reconstructions.

This is an interesting piece of work that deepens our knowledge of a newly explored paleo archive (Stylasterid corals), focuses on the need for further research regarding skeletal growth and geochemical composition and presents new information on the stable isotopic composition of skeletal material. The data presented by this manuscript is of importance for communities in the fields of paleoclimate, and marine biomineralization and calcification and as such, it should be published. However, a more thorough discussion, including data from previous publications and expanding on concepts like the role of mineralogy on the reported results should be considered and included. See below for some points that can improve the strength of the manuscript.
*We noticed similar themes between both reviewers (e.g., incorporation of coral mineralogy, clarification of methods, and incorporation of previously published data) and agree that these areas need the most attention during our revisions.*

1. Introduction

A more extensive literature review on stylasterid corals, and more specifically previous geochemical publications of this taxon and its positive results for reconstructions, would help making a stronger case on why keep focusing efforts on these specimens. This can be included either towards the end of the third paragraph or in the fourth.
*We agree and will add this section.*

2. Methods

L. 96: It is unclear from which specific dredging the two samples come from, or whether dredging D05 to D09 was done consecutively and there is no possible way to know the exact depth of the sample. This needs to be specified.

*Yes, as written, it is not clear. The dredges were consecutive, but it is unclear to us where exactly the corals were collected (from dredge D05, D06, D07, D08, or D09). This is because we performed the geochemical analyses after receiving the corals from a now deceased collaborator and we have not been able to locate sample collection information recorded during that cruise. We will clarify the methods.*

3. Results

A table in results summarising average d18O and d13C from each sample (or section as described by the text) coupled to the average environmental data used for both samples (Temperature, d18Osw and d13Csw) would help the reader to quickly grasp variability (or lack of) of the data between samples and the environmental conditions. I might be wrong, but I think seawater temperature is not specified before Figure 9 (the very end of the MS) and would help contextualise the environment and the discussion later on when comparing with work in the literature if included in the results.

*We agree that adding a table would be an easy way for potential readers to make those comparisons and will include one. The data are compiled in the supplemental information, but we will provide averages as suggested by the reviewer. This reviewer is also correct in that the seawater temperature is not mentioned before Figure 9 and we will be sure to include that in the section about sample collection as well.*

4. Discussion

4.1. Isotopic disequilibrium

This is a nice section that sets the argument for consecutive discussion on growth models and paleo reconstructions. However, I feel that a deeper comparison of the data of this manuscript with published d18O and d13C from stylasterid corals needs to be addressed (beyond the minimum offsets from equilibrium and d13C–d18O slopes). Samples here show a wider range of both d18O and d13C than those in Samperiz et al. (2020) and Wisshak et al. (2009) for aragonitic specimens. Importantly, the d13C here reach levels similar to those of calcitic samples. Whether this is an effect of much finer sampling,  mixed mineralogy, or other potential artifacts, it needs to be discussed more deeply in this section.

*We agree and will be enhancing the discussion section as per this reviewer's and the first reviewer's comments. We plan to include previously published data (from both Samperiz et al. (2020) and Wisshak et al. (2009)) into Figure 4 and Figure 5. In addition to this we will be incorporating a discussion on mineralogy as well.*

L. 205: What is the lifespan of stylasterid corals? This information can be added either here or in the introduction. For reference, observed axial growth rates from King et al. (2018), Miller et al. (2004), and/or Wisshak et al. (2009) would extrapolate to lifespans of >100yr (and up to 400yr) for Errina sp. colonies. Despite this method presenting several caveats, it is useful for the reader to understand that these colonies can be long-lived.

*We agree and will add that information.*

L. 225: Maybe I misunderstood. 0.68‰ and 3.95‰ are offset values from calcite or aragonite equilibrium? It doesn't change the observation but would be good to specify. Especially since the comparisons with other data (Samperiz et al. (2020) and Wisshak et al. (2009)) are made for aragonite equilibrium, and they show that calcitic samples show a larger offset from equilibrium for d13C.
*The offset values are from calcite equilibrium. We agree that this should be clarified and will adjust the text to be sure the proper comparisons are made.*

L.226: The minimum offset from Samperiz et al. (2020) is from bulk sampling or cross-section analysis within one specimen (similar to this study?). Worth being specific here.
*These values were compared to the cross-section analysis, but we will specify.*

L. 231: Authors comment that they have evidence for mixed mineralogy of these specimens. What is this evidence? This needs to be expanded. Work by Samperiz et al. (2020) and Stewart et al. (2020, 2022) have shown how sample mineralogy have a great impact on elemental and isotopic composition. This sentence is the only mention to mineralogy in the manuscript; however, it is known how sample mineralogy is one of the main caveats to the use of stylasterid specimens. A more thorough discussion needs to be included on the potential effects of mineralogy on these results, especially if mixed CaCO3 polymorphs have been observed. This is an interesting point that needs to be considered and will enrich the manuscript.
*We completely agree here. We have some mineralogical data, but it was not generated from the specimens that were analyzed for stable isotopes, so it was excluded. However, the samples analyzed for mineralogy were from the same cruise and dredge(s). After reading these comments, and similar comments from another reviewer, we feel comfortable including the data we have. We will edit the discussion to include the mineralogical data and discuss how that affects stable isotopes. We do have evidence for a mineralogical change in percentage of calcite-to-aragonite from the outer portion to the center. In adding to the discussion, we are going to include an additional author, Dr. Noel P. James (Queen's University), who conducted these analyses. We have also done some preliminary calculations based on the small change in percent calcite-to-aragonite by region (~3% more aragonite in the interior compared to outer samples of coral slices). This change in mineralogy could only explain the isotope values we observe if the oxygen isotope ratio of the new calcite was ~-32.5‰. This seems unrealistic. Therefore, with the caveat that the mineralogical data are from a different specimen, we will discuss that mineralogy is an important consideration, however, not the main forcing of the trends we observe. Only a small amount more of new calcite would have to be precipitated to bring isotope values to within our observed skeletal variability, but such proportions would not be parsimonious with our mineralogic data.*

4.2. Isotopic trends and calcification models

I enjoyed reading through this section. It is clear that new research needs to be directed towards modelling of stylasterid growth to clarify vital effects patterns. The addition of a simplified model with three scenarios (regular growth, ontogenic decrease and increase of radial growth) is helpful for the reader.
*Thank you!*

L. 256: The authors use published radial growth rates from Bamboo corals in their models, justified by the lack of data on growth rates from Errina sp. (or even any Stylasteridae coral). Assuming the authors still have access to the samples dated in King et al. 2018, would it be possible to roughly extrapolate radial growth across two dated points within a branch and differences in diameter? This data, although rough, could shed light on whether growth rates of Stylasteridae are similar to those employed here.

This information, coupled to lifespan of the colonies (see above, L. 205) will be useful and interesting for future research.

*This is a great suggestion to get a radial growth rate for these corals. We will include a calculation of radial growth based on the distance between radiocarbon dates and the radius of the branch.*

L. 274-275 "…as the available literature supports slowing growth with time.": Can you specify the literature (add references)? Is this referring to Stylasteridae or marine calcifiers in general?
*Yes, we will add those references. The statement was referring to marine calcifiers in general.*

L. 300 and below: Building the growth of Errina sp. on growth models from scleractinian corals (or Acroporids) can be problematic. Scleractinians have shown to calcify from centres of calcification/amorphous crystals/ fusiform crystals, from where aragonite needle-like bundles grow (e.g., Gladfeiter 1982). In Scleractinia, these calcification areas show distinct isotopic signature (e.g., Adkins et al. 2003). However, these centres of calcification or growth framework has not been observed in Stylasteridae corals and therefore is hard to argue they are the cause for isotopic differentiation.
*Whereas the calcification described by Adkins et al. (2003) applies to the solitary coral Desmophyllum cristagalli, the calcification described by Gladfelter (1982) is for a branching scleractinian coral. From our understanding, the centers of calcification in the D. cristagalli are different from those of the A. cervicornus. We don't argue that there is the exact same calcification mechanisms, only that there is a change in calcification rate: faster near the outer edges and slower near the center. We can adjust the wording to be clearer in this section.*

Wisshak et al. (2009) discussed skeletal architecture and skeletal reorganisation and it is an important source to cite and consider when studying structural growth of Errina sp. This work needs to be included in this section. Wisshak et al. (2009) explain structural growth of Errina dabneyi based on a 2-step model also, with the coenosarc canal network in the middle area of branches being simultaneously dissolved (wider-canals) and infilled (secondary precipitation) as the skeleton thickens. Although several questions remain on the nature of this secondary material, this growth model needs to be considered in this section of the discussion.
*We agree and will incorporate this into the section.*

Is there imaging showing the two-step infilling process described in this manuscript? SEM images similar to those in Figure 6 from Wisshak et al. (2009) would be helpful to discern the two-step growth. Maybe observations made during the SEM analysis. But white-light images could be useful too. Wisshak et al. (2009) describe how ampullae are more common in the outer layers, while old ampullae towards the centres of branches could be seen infilled. By the pictures of Figure 3 it would seem like that is the case (no ampullae in the inner sections), but a closer inspection could be beneficial. Just a few sentences signalling whether any of the observations made by Wisshak et al. (2009) are visible on these specimens (or not at all) will be valuable information contributing to the understanding of Errina fissurata growth.
*We agree with this too and have made arrangements to take some SEM images looking for such features. There are a few coral slices that appear to have more ampullae in the outer layers, but we will take more comprehensive images and incorporate these results into the updated manuscript.*

L. 304 "We posit that this model accurately described the stylasterid coral growth…". Disagree. This growth (outer framework and later infilling of the centre) does not explain observations by Samperiz et al. (2020) and Wisshak et al. (2009), therefore stylasterid coral growth is still largely unknown (L. 311).

*We will reword this section to clarify that we posit the growth model could apply to the specimens in this study specifically.*

An expanded discussion of sample mineralogy (as specified above, L. 231) coupled to the two-step growth model will be beneficial in here (as a paragraph or a new section within the discussion by itself). While Samperiz et al. (2020) and Wisshak et al. (2009) confirm their Errina antarctica and Errina dabneyi samples are 100% aragonite, authors hint at a mixed mineralogy here. I appreciate mineralogical mapping (e.g., Raman) or even bulk XRD might not be possible for these specimens of Errina fissurata in this manuscript. However, the possibility of the centre infilling to be mineralogically distinct from the initial framework needs to be considered as a source of discrepancies between results in this study and others published, gaining a more thorough discussion on the growth of Errina sp. Both Samperiz et al. (2020) and Stewart et al. (2020 and 2022) noted geochemical differences among calcitic and aragonitic Stylasteridae.

*We agree. As mentioned above, we have mineralogical data that we will incorporate. This consists of some XRD measurements in the center and outer region of the corals. We will include these data and expand the discussion to include the mineralogy and SEM discussions. See replies for L. 231 above for more details.*

4.3. Considerations for paleoceanographic reconstructions

L. 314: Please, specify that the white centre is the ideal region to sample for paleotemperature reconstructions "in the samples of this study". Other samples in the literature show the opposite behaviour, and therefore this cannot be extrapolated to every Errina sp. specimen. This might be species specific effect, or site-specific, or even specimen-specific.

*Yes, we will do this. This comment echoes those from another reviewer on the common theme that we need to specify this study.*

L. 338: "If finer-scale samples were informed with CT scanning methods…". Maybe I have missed it, but it is not clear to me what finer structures I should look for in the CT images to improve reconstructions. Is this denser or lighter skeleton because it would be an indication or more or less secondary infilling of the initial framework? A sentence here clarifying would be useful to guide future work.

*The CT scanning methods could illuminate structures like growth rings that are invisible to the naked eye or microscope. Additionally, the density differences between calcite and aragonite could be determined spatially. We will mention this and discuss it within the context of the mineralogy work that will be incorporated into the updated manuscript.*

L. 342: "We recommend sampling of the white centre using more spatially precise micro-milling methods…". As mentioned above, sampling the white centre would work for these specimens, but not for other published data. In addition, the white centre limits the application of this technique to specimens showing distinct coloration on its cross-section. As an example, the coenostum of Errina dabneyi sampled by Wisshak et al. (2009) was pure white, potentially not showing a distinct branch centre. I would be very precise specifying that this technique cannot be universally applied to every Errina sp. However, a fine spatial analysis on a cross-section will be useful to inform on isotopic distribution of new samples, regardless of skeletal coloration. In my opinion this is a very important point that this manuscript raises.

*We agree and will improve the section so that we are very specific about applying this technique to this species of coral and our recommended sampling scheme is not universal.*

L. 355: I would also suggest including literature of CT imaging of stylasterid corals (Stylaster sp.) showing skeletal structures (e.g., Puce et al. 2011). We know that these are structurally very different from Corallum sp., and it is not certain they follow the same growth pattern. Furthermore, skeletal structure seems to differ even between Stylasteridae genera. CT imaging will be very useful to discern growth patterns before reconstructing temperatures indeed.

*Thank you for this recommendation, we will incorporate this into the discussion.*

L. 356: would improve "and?" allow an even closer approach…

Great catch, we will fix this line.

5. Conclusions

L. 380: Please, change "we recommend sampling along the centre, white region where the infilling has allowed for calcification closest to seawater equilibrium…". In my opinion the evident recommendation emanating from this study is the need for spatial sampling to localise the skeletal region closest to equilibrium, in contrast to what was proposed for example in Samperiz et al. (2020) or Stewart et al. (2020) (i.e., bulk or surface sampling).

*We agree and will adjust the text.*

Figure 4: Add circles and squares for data from each specimen of this study (similar to Figure 5 and 9). This will help quickly localise differences across samples (or lack of thereof). Equally, including in this figure data from Samperiz et al. (2020) and Wisshak et al. (2009) would be beneficial to framework what is discussed in this section (i.e., the offset from equilibrium, differences in isotopic signal across literature sources and what might be caused by).

*We agree and will add the data and make the adjustments to the figure.*

Adkins JF, Boyle EA, Curry WB, Lutringer A (2003) Stable isotopes in deep-sea corals and a new mechanism for "vital effects." Geochim Cosmochim Acta 67:1129–1143

Gladfeiter EH (1982) Skeletal development in Acropora cervicornis: I. Patterns of calcium carbonate accretion in the axial corallite. Coral Reefs 1:45–51

Miller K, Mundy CN, Lindsay Chadderton W (2004) Ecological and genetic evidence of the vulnerability of shallow-water populations of the stylasterid hydrocoral Errina novaezelandiae in New Zealand's fiords. Aquat Conserv Mar Freshw Ecosyst 14:75–94

King TM, Rosenheim BE, Post AL, Gabris T, Burt T, Domack EW (2018) Large-Scale Intrusion of Circumpolar Deep Water on Antarctic Margin Recorded by Stylasterid Corals. Paleoceanogr Paleoclimatology 33:1306–1321

Puce S, Pica D, Mancini L, Brun F, Peverelli A, Bavestrello G (2011) Three-dimensional analysis of the canal network of an Indonesian Stylaster (Cnidaria, Hydrozoa, Stylasteridae) by means of X-ray computed microtomography. Zoomorphology 130:85–95

Samperiz A, Robinson LF, Stewart JA, Strawson I, Leng MJ, Rosenheim BE, Ciscato ER, Hendry KR, Santodomingo N (2020) Stylasterid corals: A new paleotemperature archive. Earth Planet Sci Lett 545:116407

Stewart JA, Robinson LF, Day RD, Strawson I, Burke A, Rae JWB, Spooner PT, Samperiz A, Etnoyer PJ, Williams B, Paytan A, Leng MJ, Häussermann V, Wickes LN, Bratt R, Pryer H (2020) Refining trace metal temperature proxies in cold-water scleractinian and stylasterid corals. Earth Planet Sci Lett 545:116412

Stewart JA, Strawson I, Kershaw J, Robinson LF (2022) Stylasterid corals build aragonite skeletons in undersaturated water despite low pH at the site of calcification. Sci Rep 12:13105

---

## Referee Report (RR1)

*These comments refer to the new uploaded manuscript (rather than the tracked changes file). I noted some discrepancies between the tracked changes and the revised manuscript, and I assumed (maybe incorrectly) that the tracked changes file was not a final version.*

After an initial review, some of the comments and concerns regarding this manuscript have successfully been addressed. The introduction is now more informative on stylasterid corals and the missing (but crucial) data on specimen mineralogy has been included.

I still think this data is of high value for the scientific community, and as such it should be published. However, the discussion of this manuscript (MS) is still lacking depth and details when it comes to differences between this manuscript and previously published data which is the most important and interesting perspective this new data (and MS) provides.

**Introduction**

There is no mention of mineralogy within stylasterid corals in the introduction. It is hard for a non-expert reader to then understand why mineralogy is tested, and the significance of mixed mineralogy discussed later in the manuscript.

**Line 41:** fidelity.

**Line 74 "Samperiz et al. (2020) noted variability among the δ18O […] within a single growth band."** This statement is not accurate. Samperiz et al. (2020) noted a variability of <0.50‰ for δ18O within a single growth band. Unlike δ13C (variability of ~3‰). This statement seems to point that variability of both δ18O and δ13C within a single growth band was similar. Need to be precise.

**Methods**

**Line 116:** Here you state that that species level identification was achieved through SEM imaging. Later, in Line 433 you state that the samples for this paper were not analysed using SEM, but other specimens were (I assume that this other specimen is where Figure S3 comes from). Need to specify this in methods. Are the samples analysed for taxonomical information from the same dredges/locations? And the sample in Figure S3? These details are important.

Equally, it is necessary to identify the sample ID from where SEM was obtained for future references, reproducible science, and transparency. The same applies for any other sample analysed in this manuscript that are not EA-11 and EA12 (i.e., what is the sample ID of specimens analysed for mineralogy?).

**Line 154 (?):** Do you apply a calcite-acid fractionation factor or an aragonite-acid fractionation factor to gas δ18O data? This information is missing, and it is important when considering offsets from equilibrium in Figure 6 (extended in my comment below about Line 384).

**Discussion**

In **line 339 "our results do not directly support those of Samperiz et al. (2020)"; line 361 "These results neither completely support nor refute those of Samperiz et al. (2020)"**. These two lines are within the same paragraph, yet they point towards different directions.

Upon reading further, the reader then realises that because of sampling differences (sampling growing tips v. slices 1cm below the growing tip), comparing results of this manuscript and those of Samperiz et al. (2020) is difficult. This is correctly stated by the authors in line 371, at the end of the paragraph. Therefore, it is very hard to affirm that "results do not directly support those of Samperiz et al. (2020)" (as in line 339).

This comparison, and discussion on internal variability along the growth axis is necessary and definitely enriches this manuscript. For example, the fact that maximum $\delta18O$ and $\delta13C$ in EA-12 are found in different slices is an interesting observation. However, this paragraph is long, and has contrasting statements. It is not until the very end that one understands that the comparison with previously published data (i.e., Samperiz et al. 2020) is not straightforward. This need to be re-written for accuracy, acknowledging the limitations about the comparison and being very precise about similitudes and differences in both sampling and data.

**Line 357**: Is this meant to be EA-12a?

**Line 383 "Samperiz et al. (2020) observed that calcitic specimens exhibited $\delta18O$ and $\delta13C$ values further from equilibrium compared to aragonitic corals"**: As per Samperiz et al. 2020 data (and their Figure 3), it would seem that calcitic stylasterids show more depleted values for both $\delta18O$ and $\delta13C$, but when considering equilibrium (because it is different for aragonite and calcite), calcitic specimens are further away from $\delta13C$ equilibrium but not for $\delta18O$. Modify this statement.

**Line 384 and below "Because our calculated equilibrium $\delta18O$ and $\delta13C$ values were higher for aragonite than calcite, our data relative to aragonite equilibrium [...]"**. When calculating $\delta18O_{equilibrium}$ – $\delta18O_{coral}$, it seems clear that you calculate equilibrium for calcite and aragonite (as it is different). However, it is not clear to me whether you apply different acid-fractionation corrections to your coral data depending on whether you considered it calcite or aragonite. It is my understanding that it would only make sense to compare data against aragonite equilibrium if this has been corrected with an aragonite-acid fractionation factor, and with calcite equilibrium if this has been applied a calcite-acid fractionation factor. I have mentioned this also in the methods, where it should be specified.

Equally, throughout the manuscript, and given mineralogical results, it will be more appropriate to use calcite equilibrium (as samples seem to be mostly calcitic).

**Line 388 "Our data are distinguished when we examine the heterogeneous $\delta18O$ and $\delta13C$ values across a coral slice"**. I am not sure what this sentence means. Also, not sure how it fits with what is being discussed in this paragraph.

**Line 395 "This contradicts results of Stewart et al. (2020) wherein authors find that stylasterid corals of mixed mineralogy exhibit less variability compared to purely aragonitic scleractinians and stylasterids"**. In section 3.2 of Stewart et al. (2020) the authors analyze internal variability (heterogeneity) within coral samples by calculating the difference between replicates (Table 2 of their paper). The authors only report mean differences for $\delta18O$, not $\delta13C$, so this statement cannot be hold for both stable isotopes. Nevertheless, the mean difference in $\delta18O$ replicates for aragonitic stylasterids is 0.14‰ (for n=16), and for mixed mineralogy stylasterids is 0.13‰ (for n=4). It will be hard to argue that this difference between mineralogies is significant and mixed are less variable without carrying out further statistical analyses. Change the written statement from "less variability" to "similar variability" in line 396 so that it correctly addresses data presented by Stewart et al. (2020). The fact that data in this manuscript contrast with previously published data remains true, yet the differences in sampling and analysing heterogeneity should be addressed here.

**Line 397 "[…] but they are characterized by the largest stable isotope variability."**. This sentence is confusing, and it is not clear what the authors are pointing at. Re-write for clarity.

**Line 407**. References of data of these corals (E. dabneyi, E. antarctica and bamboo corals) needed.

**Line 408 "across their surfaces, perpendicular to the growth axis"**. This seems confusing. Is it across the surface of the coral (i.e., the outer material in contact with seawater)? Is it the surface of each cross-sectional slice? Change wording.

**Line 410 "external corals".** I am assuming this refers to previously published data of E. dabneyi, E. antarctica and bamboo corals but it is not clear. Re-write for clarity.

**Line 410.** In the last sentence of this paragraphs the authors explain why their data is different to previously published data. I am not sure why there is still no mention of the role that changing mineralogy might be playing here. These MS is the first published data of highly resolved stable isotopes in mixed mineralogy stylasterids, while previous published data is on specimens either 100% calcitic or aragonitic. I understand the authors are defending that it is a growth pattern and not a mineralogy effect. Yet, I think adding a statement clarifying that these differences exist and referring to section 4.2.3 on to why mixed mineralogy is not the source of isotopic variability would help structuring the narrative and showing that mineralogy has really being considered.

**Line 417 "The presence of symbionts could contribute metabolic carbon and/or oxygen to the pool from which corals calcify"**. References for this statement.

**Line 418 and 419 "Depending on the relationship […] (Epstein et al. 1951)" -** *Carbonate-water isotopic temperature scale*, Geol. Soc. Am. Bull., 62, 417–426: I might be wrong, but I don't think this statement (symbionts near calcification sites can shift isotopic signal to lower values) emanates from the cited paper.

**Line 435:** What sample this SEM comes from (Sample ID). The specimen used for SEM was collected alive or dead? Equally, what part of the coral is Figure S3 from (i.e. main trunk without visible polyps/cyclosystems, growing branch with high presence of cyclosystems?, center of a slice or towards the outer layer?). Information on the context is missing that can be important to interpret the data provided.

**Line 443 about single coral variability as per Stewart et al. (2020)**: See comment above.

**Line 447 "[…] but corals from the same dredges were."** As above, details and information on sample ID for this analysis is missing.

**Line 448 and 4.2.3 in general**: Authors mentioned the use of false-colour imaging (Figure S4). This is the first time this method is mentioned in the MS. This information needs to be included in the methods section.

Is this false-colour image obtained via EDS, Raman, or any other method? Furthermore, this false-colour imaging was done on an alive or dead specimen? What was the sample ID? Is the same sample as one analysed via XRD or another third sample? There are VERY important information missing from this manuscript.

Importantly, mineralogy data from the false-colour imaging differs from the one obtained via XRD. While data from XRD shows an up to 5% difference between center and external material, in the false-color image this difference is of up to 18%. The authors base their argument that mixed mineralogy is not the cause for the observed discrepancies with published data (Wisshak et al. 2009 and Samperiz

et al. 2020 – both 100% aragonite) due to the small mineralogical change (<5%). Considering Figure S4, and not knowing from what sample this data is coming, it seems reasonable to question the possibility that mixed mineralogy percentages might not be similar across a given *E. fissurata* population. And if this is the case, one can question that EA-11 and EA-12 might as well have a higher percentage difference between the inner and the outer slice than those analysed via XRD. Ideally, XRD would be done on EA-11 and EA-12. But if this is not possible at all, the limitations of these observations need to be discussed together for their implications in interpreting the results.

Furthermore, it would be interesting to include what method was followed in Figure S4 to separate outer and central. Is it white versus colored coenustum? Or a random circumference of radius 1cm? This is very important to be able to compare with stable isotopic data.

This manuscript feels like mineralogy effects, and the fact that these specimens show a mixed mineralogy (although likely higher calcite than aragonite) are only visited briefly and without paying much attention to it. I would like to see a deeper discussion on mineralogy, rather than rapidly brush it off within one sentence. Importantly, this point does not mean the authors are wrong about differences in growth patterns for *E. fissurata* (with higher vital effects in the outer regions of a cross-section). As growth model of stylasterids are largely unknown, it very well could be that some populations not only show differences in mineralogy, but also in the entire growth strategy they follow. These results are fascinating, and really indicate how much more we need to learn about this taxa, but needs a deeper discussion so non-expert readers can grasp the importance of the taxa and further research.

Furthermore, whenever the temperature calibration from Samperiz et al. (2020) is used across the manuscript, at the very least it should be noted that this calibration was obtained for 100% aragonitic specimens and that it is unknown whether it is applicable for calcitic (or mixed) specimens, or that a δ18O – temperature calibration for calcitic specimens could have a different slope hence the need to be cautious. I understand that due to the lack of a better fitted calibration, Samperiz et al. (2020) is used, but it is good to be clear about limitations here.

**Section 4.3 Hypothesizes large-scale calcification model for *E. fissurata*:** This is just a suggestion and I leave the decision to modify it to the editor/authors of the study.

This section feels very long, and somewhat confusing. It goes for two long paragraphs on growth models of other coral taxa (Corallium and acroporids) to, just at the very end, conclude that "we do not suggest that the same crystal structures are apparent here. We instead hypothesize […]". I think this section would benefit of being more succinct and straight to the point. Furthermore, the growth model for stylasterids proposed by Wisshak et al. (2009) is only briefly mentioned. Reconciling or comparing this previously published model, with the authors data (both mineralogy and stable isotopic data) would be a valuable insight. Instead, it is really focused on other corals taxa for no apparent reason.

Conclusions

There is no mention of mineralogy and its role in the conclusions of the manuscript. As stated in my comment about section 4.2.3, the discussion about the role of mineralogy needs to be deeper and more detailed. As such this should also be reflected in the conclusions.

**Figure 5 and 6**: What data is from what sample? Add circles and squares (Like in Figure 4 and 7) to differentiate between data from EA-11 and EA-12.

---

## Editor Decision (ED1)

[revised manuscript text omitted]

[The linear regression is build upon what? Isotopes vs depth or vs. d13C? The isotopic range is quite wide, so this needs more clarification]

**Table 2:** Summary data for *E. fissurata* and regression statistics for the resulting compilation (see Fig. 4).

**3.1 Stable carbon and oxygen isotope trends**

Each coral slice exhibits a wide range of stable carbon and oxygen isotope ratios that are higher in the center, white section, and lower in the outer pink section of the coral (Fig. 3; Table S1 ). For specimen EA-11, the ranges in $\delta^{18}O$

190 and δ¹³C values across slices EA-11a through EA-11c are ~1.5 ‰ and ~5.2 ‰, respectively. This variability is too large to produce accurate paleoceanographic records without knowledge of where the most accurate, or closest to that of an environmental signal, is preserved in the coral skeleton. The slightly larger range of δ¹⁸O and δ¹³C values observed over EA-11d (2.2 ‰ and 6.0 ‰, respectively) is not as surprising as it could be due to the sampling scheme for this slice covering a larger vertical distance of coral than the others (i.e., transecting more time intervals). The compilation of EA-11 δ¹⁸O values

[Figure]

**Figure 3:** Stable isotope ratio maps for slices from specimens EA-11 and EA-12. Each horizontal pair of images show data from the same coral slice: the left column depicts $\delta^{18}$O values and the right column $\delta^{13}$C values. The measurements are presented as colored circles on each slice with corresponding adjacent color bar. Values are expressed in per mil (‰) relative to PDB, and analytical uncertainty (1σ) during this study was ±0.064 ‰ $\delta^{18}$O and ±0.083 ‰ $\delta^{13}$C. All slices exhibit the same feature of the highest isotopic values toward the inner white section, which is not always the geometric center of the slice.

[Figure]

205    ranges from 0.65 ‰ to 2.87 ‰, and the $\delta^{13}$C values range from -9.56 ‰ to -2.64 ‰, with the minimum values exhibited by

EA-11a (nearest the tip) and maximum values by EA-11d (far from the tip; Table 2). Although this trend is not surprising (see

Samperiz et al., 2020), if we consider "bulk" values that would be obtained by more traditional coral drilling methods, which

average isotopic compositions by drilling into the side of a coral, there is no significant change in either isotopic composition

up-coral (Fig. 4).

210

[Figure]

I would recommend saying average than bulk, as "average" best reflect what you present here and not a bulk analysis

This figure is referred in Table 2 above, but I struggle to understand how the linear relationship was calculated.

[revised manuscript text omitted]

refer to suppl., and if possible show the other figures of EA-22 and 23 instead onf EA 24 alone, and indicate

specimens EA-22, EA-23, and EA-24 for physical signs of diagenesis. Although we did not image specimens EA-11 or EA-whereabout in the sample this image was taken. The scale for SEM and the whole sample is different

[revised manuscript text omitted]

---

## Author Response (AR2)

**Review #1**

*We thank the anonymous reviewer for their constructive comments and thorough review of this manuscript. Below are the reviewer's comments and author responses in blue italics.*

These comments refer to the new uploaded manuscript (rather than the tracked changes file). I noted some discrepancies between the tracked changes and the revised manuscript, and I assumed (maybe incorrectly) that the tracked changes file was not a final version.
*This was a correct assumption, apologies for the confusion.*

After an initial review, some of the comments and concerns regarding this manuscript have successfully been addressed. The introduction is now more informative on stylasterid corals and the missing (but crucial) data on specimen mineralogy has been included.
*We have noticed common themes between both reviewers' comments in this round of revision. One of them is the need for improved discussion of mineralogy. We have taken these comments into serious consideration and have thoroughly edited the introduction, results, discussion, and conclusion sections to all include mineralogy.*

I still think this data is of high value for the scientific community, and as such it should be published. However, the discussion of this manuscript (MS) is still lacking depth and details when it comes to differences between this manuscript and previously published data which is the most important and interesting perspective this new data (and MS) provides.
*Thank you for this perspective. We have spent considerable time reworking the discussion of this manuscript to highlight the differences between what we are presenting and previous work.*

**Introduction**

There is no mention of mineralogy within stylasterid corals in the introduction. It is hard for a non-expert reader to then understand why mineralogy is tested, and the significance of mixed mineralogy discussed later in the manuscript.
*We agree that the importance of mineralogy was underrepresented throughout this manuscript and the introduction has been reformatted to include its significance.*

**Line 41:** fidelity.
*Done*

**Line 74 "Samperiz et al. (2020) noted variability among the δ18O […] within a single growth band."** This statement is not accurate. Samperiz et al. (2020) noted a variability of <0.50‰ for δ18O within a single growth band. Unlike δ13C (variability of ~3‰). This statement seems to point that variability of both δ18O and δ13C within a single growth band was similar. Need to be precise.
*We see how this statement is misleading and will take care to use more precise wording for the next submission of this manuscript. We have reworked the introduction, and this statement has been removed entirely.*

**Methods**

**Line 116:** Here you state that that species level identification was achieved through SEM imaging. Later, in Line 433 you state that the samples for this paper were not analysed using SEM, but other specimens were (I assume that this other specimen is where Figure S3 comes from). Need to specify this in methods. Are the samples analysed for taxonomical information from the same dredges/locations? And the sample in Figure S3? These details are important.

*We agree, this was unclear as written. This has been updated to specify which samples were analyzed using SEM.*

Equally, it is necessary to identify the sample ID from where SEM was obtained for future references, reproducible science, and transparency. The same applies for any other sample analysed in this manuscript that are not EA-11 and EA12 (i.e., what is the sample ID of specimens analysed for mineralogy?).

*Thank you for this comment, as you have made clear the importance of transparency when describing methods exactly. It was an oversight to not have included sample names for each step, and they have been done now.*

**Line 154 (?):** Do you apply a calcite-acid fractionation factor or an aragonite-acid fractionation factor to gas δ18O data? This information is missing, and it is important when considering offsets from equilibrium in Figure 6 (extended in my comment below about Line 384).

*Please see our response below to this reviewer's comment about Line 384.*

**Discussion**

In **line 339 "our results do not directly support those of Samperiz et al. (2020)"; line 361 "These results neither completely support nor refute those of Samperiz et al. (2020)"**. These two lines are within the same paragraph, yet they point towards different directions. Upon reading further, the reader then realises that because of sampling differences (sampling growing tips v. slices 1cm below the growing tip), comparing results of this manuscript and those of Samperiz et al. (2020) is difficult. This is correctly stated by the authors in line 371, at the end of the paragraph. Therefore, it is very hard to affirm that "results do not directly support those of Samperiz et al. (2020)" (as in line 339).

*This section was unclear as written and has been reworked, see further comments below.*

This comparison, and discussion on internal variability along the growth axis is necessary and definitely enriches this manuscript. For example, the fact that maximum δ18O and δ13C in EA-12 are found in different slices is an interesting observation. However, this paragraph is long, and has contrasting statements. It is not until the very end that one understands that the comparison with previously published data (i.e., Samperiz et al. 2020) is not straightforward. This need to be re-written for accuracy, acknowledging the limitations about the comparison and being very precise about similitudes and differences in both sampling and data.

*We agree that this paragraph meanders and we contradict ourselves several times. The beginning was a poor choice of words to try and develop the nuance in the differences between our results and those of Samperiz et al. (2020). We have reformatted the discussion section and have clarified the comparisons with other works. We have also adjusted this part of the discussion to more*

*clearly state limitations introduced by sampling strategy. This new structure flows much better for the reader, and we appreciate these constructive comments.*

**Line 357**: Is this meant to be EA-12a?
*Yes, thank you for the catch, this has been changed.*

**Line 383 "Samperiz et al. (2020) observed that calcitic specimens exhibited δ18O and δ13C values further from equilibrium compared to aragonitic corals"**: As per Samperiz et al. 2020 data (and their Figure 3), it would seem that calcitic stylasterids show more depleted values for both δ18O and δ13C, but when considering equilibrium (because it is different for aragonite and calcite), calcitic specimens are further away from δ13C equilibrium but not for δ18O. Modify this statement.
*Yes, we agree that the wording was misleading regarding the isotope values relative to equilibrium. In editing this section, this sentence has been removed.*

**Line 384 and below "Because our calculated equilibrium δ18O and δ13C values were higher for aragonite than calcite, our data relative to aragonite equilibrium […]"**. When calculating δ18Oequilibrium − δ18Ocoral , it seems clear that you calculate equilibrium for calcite and aragonite (as it is different). However, it is not clear to me whether you apply different acid-fractionation corrections to your coral data depending on whether you considered it calcite or aragonite. It is my understanding that it would only make sense to compare data against aragonite equilibrium if this has been corrected with an aragonite-acid fractionation factor, and with calcite equilibrium if this has been applied a calcite-acid fractionation factor. I have mentioned this also in the methods, where it should be specified.
*We thank the reviewer for this comment as it was very illuminating and interesting to read about. We agree that the acid fractionation factor (AFF) we used should be clearly stated. In our study, the acid fractionation was accounted for by measuring calcite reference materials, therefore the calcite fractionation factor was incorporated into data transfer onto the PDB scale and calculations based on those data. Regarding a strict division between aragonite AFF-corrected and calcite AFF-corrected values only being compared to their respective equilibria, there is more nuance. Whereas we agree that the aforementioned comparison is the most accurate method for interpreting stable oxygen isotope values, equilibrium/paleotemperature equations for both carbonate polymorphs often don't account for the AFF or simply assume calcite by calibrating with the NBS-19 standard (e.g., Grossman and Ku, 1986; Crowley, 2010). Such complications have been the topic of several correspondences among the isotope geochemistry community (see Lachniet, M., Pack, A., & van Geldern, R., Isogeochem listserv, April 2014; Gillikin, D., Isogeochem listserv, October 2020). To combat this confusion, we think that future works need to clearly state the AFF they use (or assume), previous paleotemperature calibrations for aragonite need to be revisited, and further temperature calibrations like that of Samperiz et al. (2020) need to be made wherein the AFF is accounted for.*

*Here, we consider how the different AFFs would affect the results of this study, specifically comparison to any theoretical equilibrium point in isotope space. As the acidification of carbonate with phosphoric acid only fractionates oxygen isotopes, the carbon isotopes are unaffected;*

*however, we see a strong linear relationship between the oxygen and carbon isotopes (Figure 5 in the manuscript). This suggests that the AFF only affects the absolute value of each $\delta^{18}O$ measurement on a magnitude not significant enough to disrupt the linear relationship between isotopes – perhaps adding noise to the relationship unsubstantially as the relationship is still clear. Further, calcite and aragonite AFFs determined by Kim et al. (2007) at 50°C (the temperature we acidified coral samples) would result in a difference in $\delta^{18}O$ values of ~0.03 ‰. This potential offset is two orders of magnitude smaller than the minimum range in $\delta^{18}O$ values (1.31 ‰) observed in either coral (Figure 4 in the manuscript). Such a magnitude of change can be significant for calibration to temperature, which is outside the scope of this work, but does not impact our interpretations of coral growth and the optimal location for sampling if one wanted to calibrate this species or genus to temperature.*

*References:*
*Crowley, S. F. (2010). Mineralogical and chemical composition of international carbon and oxygen isotope calibration material NBS 19, and reference materials NBS 18, IAEA-CO-1 and IAEA-CO-8. Geostandards and Geoanalytical Research, 34(2), 193-206.*

*Grossman, E. L., & Ku, T. L. (1986). Oxygen and carbon isotope fractionation in biogenic aragonite: temperature effects. Chemical Geology: Isotope Geoscience Section, 59, 59-74.*

*Kim, S. T., Mucci, A., & Taylor, B. E. (2007). Phosphoric acid fractionation factors for calcite and aragonite between 25 and 75 C: revisited. Chemical Geology, 246(3-4), 135-146.*

Equally, throughout the manuscript, and given mineralogical results, it will be more appropriate to use calcite equilibrium (as samples seem to be mostly calcitic).
*This has been done.*

**Line 388 "Our data are distinguished when we examine the heterogeneous δ18O and δ13C values across a coral slice".** I am not sure what this sentence means. Also, not sure how it fits with what is being discussed in this paragraph.
*This statement was meant to emphasize that overall, the isotopic values were within a realistic range, but were very different when considering the variability within a single slice. This was poorly written and has been removed.*

**Line 395 "This contradicts results of Stewart et al. (2020) wherein authors find that stylasterid corals of mixed mineralogy exhibit less variability compared to purely aragonitic scleractinians and stylasterids".** In section 3.2 of Stewart et al. (2020) the authors analyze internal variability (heterogeneity) within coral samples by calculating the difference between replicates (Table 2 of their paper). The authors only report mean differences for δ18O, not δ13C, so this statement cannot be hold for both stable isotopes. Nevertheless, the mean difference in δ18O replicates for aragonitic stylasterids is 0.14‰ (for n=16), and for mixed mineralogy stylasterids is 0.13‰ (for n=4). It will be hard to argue that this difference between mineralogies is significant and mixed are less variable without carrying out further statistical analyses. Change the written statement from "less variability" to "similar variability" in line 396 so that it correctly addresses data presented by Stewart et al. (2020). The fact that data in this manuscript contrast with previously

published data remains true, yet the differences in sampling and analysing heterogeneity should be addressed here.

*We agree with this comment in that we had poorly described the variability of Stewart et al. (2020). In editing the discussion section, we removed this section because upon rereading Stewart et al., it was unclear if the mixed mineralogy specimens they present were truly mixed. The authors state that all $\delta^{18}O$ and $\delta^{13}C$ values and associated mineralogical data were from Samperiz et al. (2020). The confusion lies in the mineralogy of the E. gracilis specimens. The supplemental information provided by Samperiz et al. (2020) do not state the mineralogy of these corals, they are listed as "na". It is unclear where the mineralogical information came from for E. gracilis. Stewart et al. could have done the analyses or gotten more information from a personal communication from Samperiz et al., but it was not stated. Therefore, we chose to eliminate a reference to Stewart et al. mixed mineralogy corals.*

**Line 397 "[…] but they are characterized by the largest stable isotope variability."**. This sentence is confusing, and it is not clear what the authors are pointing at. Re-write for clarity.
*This line has been removed.*

**Line 407**. References of data of these corals (E. dabneyi, E. antarctica and bamboo corals) needed.
*This line has been removed during editing, but we will be sure to provide citations in other mentions of these works.*

 **Line 408 "across their surfaces, perpendicular to the growth axis"**. This seems confusing. Is it across the surface of the coral (i.e., the outer material in contact with seawater)? Is it the surface of each cross-sectional slice? Change wording.
*This line has been removed.*

**Line 410 "external corals".** I am assuming this refers to previously published data of E. dabneyi, E. antarctica and bamboo corals but it is not clear. Re-write for clarity.
*Yes, but this line has been removed.*

**Line 410.** In the last sentence of this paragraphs the authors explain why their data is different to previously published data. I am not sure why there is still no mention of the role that changing mineralogy might be playing here. These MS is the first published data of highly resolved stable isotopes in mixed mineralogy stylasterids, while previous published data is on specimens either 100% calcitic or aragonitic. I understand the authors are defending that it is a growth pattern and not a mineralogy effect. Yet, I think adding a statement clarifying that these differences exist and referring to section 4.2.3 on to why mixed mineralogy is not the source of isotopic variability would help structuring the narrative and showing that mineralogy has really being considered.
*Thank you for this comment. Because of the constructive comments of both reviewers, we have edited this manuscript to incorporate much more information about mineralogy. We have incorporated it into the introduction and added a much more thorough discussion section. We have also made changes to describe how mineralogy should be considered when examining the isotopic variability.*

**Line 417 "The presence of symbionts could contribute metabolic carbon and/or oxygen to the pool from which corals calcify"**. References for this statement.
*Done*

**Line 418 and 419 "Depending on the relationship […] (Epstein et al. 1951)"** - *Carbonate-water isotopic temperature scale*, Geol. Soc. Am. Bull., 62, 417–426: I might be wrong, but I don't think this statement (symbionts near calcification sites can shift isotopic signal to lower values) emanates from the cited paper.
*Good catch, this was an artifact of previous edits. It has been changed to the correct reference.*

**Line 435:** What sample this SEM comes from (Sample ID). The specimen used for SEM was collected alive or dead? Equally, what part of the coral is Figure S3 from (i.e. main trunk without visible polyps/cyclosystems, growing branch with high presence of cyclosystems?, center of a slice or towards the outer layer?). Information on the context is missing that can be important to interpret the data provided.
*Thank you for pointing out that we are missing information. This has been added to the manuscript as well as the supplemental figure captions.*

**Line 443 about single coral variability as per Stewart et al. (2020)**: See comment above.
*We understand the mistake that was made here. This section has also been heavily edited to provide more of a consideration for a mineralogical influence on the isotopic values.*

**Line 447 "[…] but corals from the same dredges were."** As above, details and information on sample ID for this analysis is missing.
*This has been edited.*

**Line 448 and 4.2.3 in general**: Authors mentioned the use of false-colour imaging (Figure S4). This is the first time this method is mentioned in the MS. This information needs to be included in the methods section. Is this false-colour image obtained via EDS, Raman, or any other method? Furthermore, this false-colour imaging was done on an alive or dead specimen? What was the sample ID? Is the same sample as one analysed via XRD or another third sample? There are VERY important information missing from this manuscript. Importantly, mineralogy data from the false-colour imaging differs from the one obtained via XRD. While data from XRD shows an up to 5% difference between center and external material, in the false-color image this difference is of up to 18%. The authors base their argument that mixed mineralogy is not the cause for the observed discrepancies with published data (Wisshak et al. 2009 and Samperiz et al. 2020 – both 100% aragonite) due to the small mineralogical change (<5%). Considering Figure S4, and not knowing from what sample this data is coming, it seems reasonable to question the possibility that mixed mineralogy percentages might not be similar across a given *E. fissurata* population. And if this is the case, one can question that EA-11 and EA-12 might as well have a higher percentage difference between the inner and the outer slice than those analysed via XRD. Ideally, XRD would be done on EA-11 and EA-12. But if this is not possible at all, the limitations of these observations need to be discussed together for their implications in interpreting the results.

Furthermore, it would be interesting to include what method was followed in Figure S4 to separate outer and central. Is it white versus colored coenustum? Or a random circumference of radius 1cm? This is very important to be able to compare with stable isotopic data.

*We appreciate this reviewer's thoughtful consideration of this section. However, we have decided to remove this figure altogether. Because we were able to perform XRD analysis on a coral piece that was also used for stable isotopes, we considered it preferable. The image processing done here is very qualitative and based on identifying the blue and red pixels in the image and separating them. They were then divided into the outer and inner portions based on the approximate scale and the average diameter of the white coral center of the other corals. This specimen was collected from the same dredge as all the other E. fissurata presented here, but the false color analysis was done years ago before it was intended to be incorporated into this work. Additional details that would be required to fully incorporate this figure into the main text are not available. Therefore, we have chosen to remove it from this manuscript.*

This manuscript feels like mineralogy effects, and the fact that these specimens show a mixed mineralogy (although likely higher calcite than aragonite) are only visited briefly and without paying much attention to it. I would like to see a deeper discussion on mineralogy, rather than rapidly brush it off within one sentence. Importantly, this point does not mean the authors are wrong about differences in growth patterns for *E. fissurata* (with higher vital effects in the outer regions of a cross- section). As growth model of stylasterids are largely unknown, it very well could be that some populations not only show differences in mineralogy, but also in the entire growth strategy they follow. These results are fascinating, and really indicate how much more we need to learn about this taxa, but needs a deeper discussion so non-expert readers can grasp the importance of the taxa and further research.

*We appreciate the constructive and supportive nature of this comment. During this revision process, in preparation to discuss mineralogy in greater detail, we have been convinced that it cannot be discounted. We agree with this reviewer and have edited this section of the discussion so that a potential mineralogy impact on the coral geochemistry will not be downplayed. This was a pleasure to research as we learned that we actually do have the first $\delta^{18}O$ and $\delta^{13}C$ records from mixed mineralogy stylasterids. It is exciting to be able to discuss these results and hopefully generate excitement and further studies into how these corals calcify.*

Furthermore, whenever the temperature calibration from Samperiz et al. (2020) is used across the manuscript, at the very least it should be noted that this calibration was obtained for 100% aragonitic specimens and that it is unknown whether it is applicable for calcitic (or mixed) specimens, or that a δ18O – temperature calibration for calcitic specimens could have a different slope hence the need to be cautious. I understand that due to the lack of a better fitted calibration, Samperiz et al. (2020) is used, but it is good to be clear about limitations here.
*We agree.*

**Section 4.3 Hypothesizes large-scale calcification model for *E. fissurata*:** This is just a suggestion and I leave the decision to modify it to the editor/authors of the study.

This section feels very long, and somewhat confusing. It goes for two long paragraphs on growth models of other coral taxa (Corallium and acroporids) to, just at the very end, conclude that "we do not suggest that the same crystal structures are apparent here. We instead hypothesize […]".

I think this section would benefit of being more succinct and straight to the point. Furthermore, the growth model for stylasterids proposed by Wisshak et al. (2009) is only briefly mentioned. Reconciling or comparing this previously published model, with the authors data (both mineralogy and stable isotopic data) would be a valuable insight. Instead, it is really focused on other corals taxa for no apparent reason.

*We agree that there are additional growth structures described that are not needed and take away from the main messages of this work. We have trimmed this section down. However, when we describe the growth model by Gladfelter (1982), we are not discounting it when we say that "we do not suggest that the same crystal structures are apparent here". The Gladfelter growth model for acroporids instead prescribes very specific details about fusiform crystals followed by aragonite crystals. We do not have the data to support if the same type of calcification is occurring on that scale, but rather the broad growth patterns could explain our isotope records. Because this was not made clear, we will edit it for clarification. Regarding the Wisshak et al. (2009) growth model, we have incorporated their description, along with other evidence for the narrow, internal mesh structure on the outside of the corals and the wider canals toward the inside of stylasterid corals.*

**Conclusions**

There is no mention of mineralogy and its role in the conclusions of the manuscript. As stated in my comment about section 4.2.3, the discussion about the role of mineralogy needs to be deeper and more detailed. As such this should also be reflected in the conclusions.

*We agree, and have edited the conclusions to include the importance of mineralogy and a call to future work developing temperature calibrations for mixed mineralogy stylasterids.*

**Figure 5 and 6**: What data is from what sample? Add circles and squares (Like in Figure 4 and 7) to differentiate between data from EA-11 and EA-12.

*We have changed the symbols of Figure 5 so that the circles represent EA-11 and the squares are EA-12. At first, it seemed as though introducing more symbols to the figure would be overwhelming, but this reviewer's perspective helped us to clarify the figure for readers. We have also changed the symbols for Figure 6 to improve understanding. We have emphasized the difference between aragonite and calcite corals and made all Errina symbols the same.*

**Review #2**

*We thank the anonymous reviewer for their constructive comments and thorough review of this manuscript. Below are the reviewer's comments and author responses in blue italics.*

**Second Review of "Deep-sea stylasterid δ18O and δ13C maps inform sampling scheme for paleotemperature reconstructions" by King et al.,**

It is good to see that many of the missing references have been included and that mineralogy has been addressed in this newest iteration of the Ms. It is a shame though that stylasterid mineralogy is still not properly addressed in the introduction, therefore the reader will be unclear as to the importance/necessity of the XRD data when they suddenly appear in the results. I still feel though that the interpretation of these mineralogical results may be too superficial, and the effects are far more important to reconciling these mixed mineralogy specimens with previous results from pure high Mg calcite and pure aragonite specimens measured before.

*We have noticed common themes between both reviewers' comments in this round of revision. One of them is the need for improved discussion of mineralogy. We have taken these comments into serious consideration and have thoroughly edited the introduction, results, discussion, and conclusion sections to all include mineralogy.*

I disagree with the interpretation that the centres are just 3% more aragonite mentioned in the response to review (the image in figure s4 clearly shows there is more aragonite than that – see below). Without direct measurement of these exact specimens (with spot sizes equal to that of the stable isotope analyses) it is difficult to rule out that there are *both* rate effects (as the authors suggest) AND mineralogical effects driving the trends.

*We thank the reviewer for this perspective. Since the last submission, we were able to run one of the coral slices for XRD that was also run for stable isotopes. This provided additional important mineralogical information which we have worked diligently to incorporate into the manuscript. We have also changed the discussion to account for the possible influence of mineralogy and have discussed mineralogical impact on skeletal isotope ratios and the potential impact of both growth and mineralogical forcing of the isotope trends.*

There are still typographic errors and mistaken figure references in text that need to be addressed. I maintain that these results are important and thought provoking and merit publication, however I still feel that the discussion needs work. My specific line by line comments are listed below.

*We thank this reviewer for their patience as we work through this manuscript. We have addressed all of these errors and have polished it as a whole.*

**Line numbers refer to the track changed manuscript pdf file.**

The abstract is still misleading. Without reading the full paper, the reader is left with the impression that stylasterid corals in general have this low to high (out to in) stable isotope distribution when the Samperiz data prior to this clearly show the opposite is true in pure

aragonite and pure calcitic stylasterids. The authors need to be clear on line 15 "growth structure of a mixed mineralogy (high-Mg calcitic and aragonite; confirmed by X-ray diffraction analysis) deep-sea stylasterid coral, Errina fissurata".

*We agree that the abstract as written would lead readers to believe that the findings we present are applicable to all stylasterids. We have edited the abstract to clarify the implications and the work done.*

Line 42. I don't think "noise" is the right term to describe vital effects. Isotope ratios could be offset from equilibrium by a consistent value of -2 ‰ for example. This wouldn't give a noisier signal, just an inaccurate one. I would recommend deleting the word "noise" and just leave it as aka "vital effects".

*We agree and have changed the wording.*

Line 63: suggest "vital effects have been invoked to explain trends in δ18O…"

*Done, thank you for the suggestion.*

Line 78. Consider rephrasing. This statement "these results obscure best practices for colony-scale sampling" implies that Samperiz results actually hindered progress towards understanding coral vital effects, when the opposite is true. Both this study and Samperiz study shed light the micro-structural complexity of stylasterids, highlighting the need for further work.

*This sentence has been removed as the introduction has been reformatted. We agree that this read improperly as it was not our intention to imply that Samperiz et al. hindered any progress.*

Line 80: one of the most interesting things about stylasterids is that they are made of aragonite, hi mg calcite or in some cases both polymorphs of carbonate. This needs to be introduced before the "Here we present.." statement as it is a key factor when interpreting difference between Samperiz work and this study.

*Yes, we agree and have done this.*

Line 83. Recent boron isotope results in Stewart et al., 2022 (Sci Reports) show that stylasterids have a fundamentally different biocalcification strategy to (well-studied) scleractinia. This study shows that they do not upregulate internal pH, so it is still unclear how stylasterids precipitate aragonite in undersaturated waters. This supports the need for further research and is useful to mention here.

*Yes, we agree and have done this.*

Line 90. If the importance of mineralogy is introduced earlier, the XRD work can be mentioned here in the "here we present" section as an important tool to help understand differences in δ18O and δ13C results between stylasterid taxa.

*We have done this as well. After incorporating this reviewer's feedback above, it flowed naturally.*

Ling 149: The importance of mineralogy needs to be laid out in the intro and mentioned in the here we present section before "2.3. Mineralogical analysis" is described. The reader needs be

clear why these analyses are needed – i.e. to establish if this is an aragonitic Errina like that of Samperiz or if there are mineralogical differences across the growth bands.

*This has been done, see responses to similar comments above.*

Line 153: this line has not been completed. "They were analyzed on a XX diffractometer with a wavelength of xxx."

*We apologize as this was an artifact of the "Tracked changes" version. The line was completed in the uploaded manuscript.*

Line 161: overuse of word "employ"

*Good catch, thank you.*

Line 225. Again, this intro to stylasterid mineralogy is too late into the Ms. This is also an oversimplification as some stylasterids are pure aragonite, some pure Hi-Mg calcite, and some are "mixed mineralogy" as mentioned here. These three modes of calcification within a coral family is strange and a key reason why we need to study these organisms and their geochemistry.

*We agree that we didn't spend enough time discussing the mineralogy of these corals and should have brought it up much sooner than this portion of the results section. We have enhanced the introduction with much more information about stylasterid mineralogy and have described the XRD results here. The stylasterid mineralogy has become a much more prominent consideration in this manuscript.*

Line 239: the statement implies results do not support that of Samperiz, then on line 245 results from specimen EA-11 do support Samperiz. This is confusing to the reader unless the work of Samperiz is clearly laid out in the introduction, stating how the new study is different, and what exactly is being tested (i.e. the first mixed mineral specimen, but same genera, sampled in a different way). The results of this study are important and thought provoking, just their place within the context of previous work hasn't been properly established.

*We agree that this was a poor choice of words to try and develop the nuance in the differences between our results and those of Samperiz et al. (2020). We have reformatted the discussion section and have clarified the comparisons with other works. This new structure flows much better for the reader.*

Line 246: Do you mean "Specimen EA-12 was characterized by δ18O and δ13C minima nearest the tip (**EA-12a**)," rather than EA-11a

*Yes, thank you for the catch, this has been changed.*

Line 253: I agree here, Figure 4 to me seems to suggest that there was effectively no difference in terms of δ18O and δ13C between samples closer to tip or base. They all have similar average and variance. The discussion above here is focused on the extreme outliers of each sample section and adds little for determining if the "tip" is lower than the main stem in its stable isotope ratio. Indeed, with no tips actually sampled in this study im not sure that this can be assessed.

*We have restructured the discussion as both reviewers have aptly pointed out its confusing nature and that this is not a direct comparison. We have changed this section to clearly state the*

*limitations introduced by the sampling strategy. We appreciate the constructive nature of these comments.*

Figure 4. Remove "analytical uncertainty" error bar from top left and just say that error was smaller than data points. At the moment it looks like the bulk average with its 2sd is the only one that had any analytical error. The legend is also poorly arranged and it is difficult to tell that there are two specimens (11 and 12) plotted here. Suggest two columns, one for EA11 and one for EA12.
*These changes have been made in addition to changing the color scheme to improve clarity.*

Figure 5. D. cristagalli is now Desmophyllum dianthus.
*Thank you for the catch. Because of this comment, we also noticed that Lophelia is now listed as unaccepted on the World Register of Marine Species and has been replaced by Desmophyllum. All mentions of the outdated nomenclature have been corrected, and we have made distinctions between D. dianthus (previously D. cristagalli and Desmophyllum sp. (previously Lophelia).*

*Additional Reference:*
*Addamo, A. M., Vertino, A., Stolarski, J., García-Jiménez, R., Taviani, M., & Machordom, A. (2016). Merging scleractinian genera: the overwhelming genetic similarity between solitary Desmophyllum and colonial Lophelia. BMC evolutionary biology, 16, 1-17.*

Figure 6. This is an important figure, but needs to be clearer. Some sort of clear distinction between published and new data (e.g. open and closed symbols), and calcites vs aragonites (e.g. blue and red symbols) and also those samples that are Errina (e.g. circles). This should clearly show that Samperiz's aragonites are all much higher δ18O and δ13C than this study. This is then a good figure to refer to when introducing the work. The caption of figure 6 also refers to a figure 6 – please check what is meant here.
*Thank you for the advice. This figure proved difficult to illustrate a clear comparison among all the stylasterid coral data. We have incorporated your suggestions and modified the figure. There is now a clear division between aragonite (empty symbols) and calcite (filled symbols) corals. I have also made each Errina coral a circle, and denoted more clearly which data points are new from this study. I have also made a distinction between specimen EA-11 and EA-12 samples using circles and squares to be consistent with the rest of the figures.*

Line 267: I do not follow the reasoning here. Samperiz found that calcites had lower stable isotope values than aragonites and were therefore further from equilibrium. That is the same as what is shown in Figure 6. Regardless of one's choice of equilibrium seawater value (arag or calcite) the blue dots from this study are all much lower in δ13C (further from equilibrium) than the 0,0 point and similar to the calcites in Samperiz. Sure, the offset is increased if an aragonite equilibrium is chosen, but I don't think that is the right approach for a predominantly calcite organism and the darker blue dots just make the figure less clear.
*These few sentences have been removed as the reviewer is correct, and it is very unclear and the incorrect approach to displaying and describing the dataset. This entire discussion has been*

*reworked and with the additional XRD analysis we were able to do, we removed any mention of the aragonite equilibrium.*

Line 278: this misrepresents the Stewart et al., 2020 study. All δ18O and δ13C values in Stewart et al., and bulk measurements come from Samperiz et al., 2020. Although Stewart et al., table 2 may give the impression that variance is smaller between replicates in mixed and hi-mg calcites, this is likely due to undersampling (only 4 and 2 specimens). For that reason, no such observation is made in the discussion section of that paper (stylasterids, in general, are merely compared to scleractinia and found to be less heterogeneous). If anything, the opposite is true when Sr/Ca and Li/Mg data are considered which show large discrepancies between replicate bulk analyses in mixed mineralogy specimens.

*We agree with this comment in that we had poorly described the variability of Stewart et al. (2020). In editing the discussion section, we removed this section because upon rereading Stewart et al., it was unclear if the mixed mineralogy specimens they present were truly mixed. The authors state that all $\delta^{18}O$ and $\delta^{13}C$ values and associated mineralogical data were from Samperiz et al. (2020). The confusion lies in the mineralogy of the E. gracilis specimens. The supplemental information provided by Samperiz et al. (2020) do not state the mineralogy of these corals, they are listed as "na". It is unclear where the mineralogical information came from for E. gracilis. Stewart et al. could have done the analyses or gotten more information from a personal communication from Samperiz et al., but it was not stated. Therefore, we chose to eliminate a reference to Stewart et al. mixed mineralogy corals.*

Line 285: D. cristagalli is now Desmophyllum dianthus.
*See response to similar comment above for Figure 5.*

Line 293: Citation needed for E dabneyi and E antarctica data here so show they are not from this study.
*This line has been removed during editing, but we will be sure to provide citations in other mentions of these works.*

Line 315: the wording in this section should be revised. These are azooxanthellate corals, therefore referring to their symbionts here is confusing. I don't think the barnacles that live on them are symbiotic, but rather epiphytes. I agree though with the premise that this is an unlikely cause of sample contamination.
*Done, thank you.*

Line 341: Again, reference to Stewart here is not valid and these selected bulk samples taken from the Samperiz dataset do not properly reflect the variance that Samperiz found between tips and bulk.
*We understand the mistake that was made here. This section has also been heavily edited to provide more of a consideration for a mineralogical influence on the isotopic values.*

Line 342. Be clear that specimens of the same species from the same dredges were measured for XRD. This all needs to come sooner and worked into the intro, methods and results sections before discussion here.

*We agree, and these edits have been made.*

Line 345. Mostly calcite, therefore a calcite equilibrium value should be used for these data in figure 6

*We agree, the aragonite equilibrium values have been removed from all figures.*

Line 347. Space needed between to and 77%

*Good catch.*

Line 349. I disagree that we can rule out that the mineralogical differences across the specimen are not the driver of high δ18O and δ13C values. Aragonites should be naturally higher in their equilibrium stable isotope values (~+2 ‰) therefore an aragonite core to the specimens would give the higher values seen in this study. Both specimens in table 3 have small increases in aragonite at the centre (+3% and +5%). I agree these are small, but I question whether the XRD sampling was at the same resolution as the spot sampling for stable isotopes. It is curious that the values given in Table 3 do not represent those shown on the image in Fig S4. This supplemental fig suggests that the inner portion of the coral is +18% more aragonite. This could be a lot more if you select a smaller circle to denote the sample inner (e.g. the green circle I have added to the figure). Without measuring the samples in the study for XRD I don't see how a predominantly aragonite core to the specimens can be ruled out as driving the data. This could be acting in conjunction with the growth rate effects suggested by the model. This is not a bad thing. This is the first example of detailed stable isotope chemistry across mixed mineralogy specimen (compared to Samperiz et al., who only measured pure calcite and pure aragonite) and it shows that perhaps a different sampling strategy is needed. Stylasterids are not as simple as we first thought and their mineralogy is very important and must be considered.

*We appreciate the constructive and supportive nature of this comment. During this revision process, in preparation to discuss mineralogy in greater detail, we have been convinced that it cannot be discounted. We agree with this reviewer and have edited this section of the discussion so that a potential mineralogy impact on the coral geochemistry will not be downplayed. This was a pleasure to research as we learned that we actually do have the first δ$^{18}$O and δ$^{13}$C records from mixed mineralogy stylasterids. It is exciting to be able to discuss these results and hopefully generate excitement and further studies into how these corals calcify.*

[Figure]

**Figure S4:** False-color image of representative coral mineralogy. The coral section is ~1cm in diameter.

|  | Original Image | Red denotes Mg (calcite) | Blue denotes Sr (aragonite) |
|---|---|---|---|

Outer portion representing pink area

79% Calcite
21% Aragonite

Inner portion representing white area

61% Calcite
39% Aragonite

I think this image needs to be included in the main text. The green circle shows a much bluer (majority aragonitic) core to the samples than is represented in the text.

*We have decided to remove this figure altogether. Because we were able to perform XRD analysis on a coral piece that was also used for stable isotopes, we considered it preferable. The image processing done here is very qualitative and based on identifying the blue and red pixels in the image and separating them. They were then divided into the outer and inner portions based on the approximate scale and the average diameter of the white coral center of the other corals. That is why the center portions are much larger than the green circle provided here. This specimen was collected from the same dredge as all the other E. fissurata presented here, but the false color analysis was done years ago before it was intended to be incorporated into this work. Additional details that would be required to fully incorporate this figure into the main text are not available. Therefore, we have chosen to remove it from this manuscript.*

Line 435. Check figure numbers. I think this section should be referring to fig 10. It would be fascinating to know if these brighter pixels denote more aragonite.
*Good catch, fixed.*

Line 477. Conclusions should also address mineralogy e.g. In the case of *E. fissurata*, a mixed mineralogy taxon, we…"
*We have included this.*

Line 477. This result contradicts growth structures hypothesized in the current body of literature based on observed stable isotopic trends that have only focused microsampling on purely high-Mg or purely aragonitic specimens.

*We have included this, thank you for the suggestion.*

---

## Author Response (AR3)

**Publish subject to minor revisions (review by editor; comments below):**

There are some minors corrections that I think you need to correct (see below), and please ensure to re-proof read your work prior to sending the final publications.
I am happy to accept this contribution for publications to Biogeosciences. Congratulations and best wishes in your future endeavors.

Voary

Author comments are noted in blue.

We would like to thank this editor for guidance and patience during the submission and review processes for this manuscript. We note that we were able to analyze an additional coral sample (as per the suggestions of previous reviews) for mineralogical analysis during review - the previous samples run for detailed mineralogical analysis were not the same specimens analyzed for stable carbon and oxygen isotopes. We now have detailed mineralogical data that support these corals analyzed here are nearly 100% calcite with an infinitesimal contribution of aragonite (<5%), rather than mixed calcite and aragonite. This does not change our interpretations of the stable isotopic data but supports our original growth/calcification hypothesis rather than a mineralogical forcing as the whole skeleton is the same mineralogy. This means that we have removed one of the tables in the manuscript (Table 3) along with mentions of mixed mineralogy and changed the supplemental Figure S3 to illustrate this. We wanted to make sure you were aware of these additions because they went above the editorial suggestions at this stage in the review process, and these additions could certainly be construed as having impact on the thrust of the manuscript. However, given that the new results confirm our original hypothesis and satisfy previous reviewers' desire to include XRD data, we felt that you would generally be pleased with these additions.

The comments below have all been addressed and the updated manuscript has been submitted.
* * *
L 49: remove of which (after latter))
Done
L50: records, such as oxygen and carbon...
Done
L214: The XRD diffraction patterns were obtained
Done
L289: remove in the supplement (the S in Table S1 is already an indication of supplementary)--I see this instance repeated in the manuscript, please correct accordingly
Done
L317: mey be extra word "up-coral", delete?
Changed the wording to "along the growth axis"
L559: in E fissurata ("in" is missing)
Done
L664: crystal instead of crustals
We could not find the instance of "crustals" in the text.
L672: there is an extra punctuation after Internal, please delete

Done

Figure formatting, Please put the figures in line with the text and not that the manuscript will surround the figures. There was some text overlap for example in Figure 10, and it is not easy to read the thin column.
Done

Note: Refer to the line numbers in the track changed version.

---

## Author Response (AR4)

Dear authors,

Following the edits and updates you have made to the manuscript, I went ahead and re-reviewed your contributions. I have suggestions for revisions, and I annotated the track-changed version of your manuscript (see attached) for your reference.

Please let me know if you have any questions, and thank you for adding more relevant dataset. Handling this manuscript was quite unusual, but we all want published manuscript to BG to be at its best version. Thanks for your perseverance.

Best wishes,
Voary

We thank the editor for another thoughtful review of this manuscript and have responded to the editor's remarks below. We have copied the comments from the track-changes version of this manuscript that was sent back to the authors. Please find the author responses in blue.

Line 120: I think the SEM method needs further details in this manuscript.
We added a few lines of extra detail here.

Line 158: Change "for mineralogical analysis" to "for mineralogy". Add here the details about the XRD on L165-167.
Done.

Line 163: If it is scanned, is the method still XRD?
We asked the analyst to clarify methods, and yes, this is still a method of XRD.

Line 168: You may need to distinguish between probe and XRD, entirely different things.
The analyst also advised us to change "probe" to analyze".

Line 181: Why is this technique suggested here? I don't follow the logic.
This technique of radiometric dating was mentioned because these corals presented an unusual internal structure, or lack thereof. This presents challenges regarding establishing chronology and determining where to accurately sample so that we progress through time. This paragraph has been reworded to clarify that intent.

Line 185: The linear regression is built upon what? Isotopes vs depth or vs. d13C? The isotopic range is quite wide, so this needs more clarification.
We have clarified the table to state the regression details ($\delta^{18}$O vs $\delta^{13}$C) and changed the caption to refer to Fig.5 instead of Fig 4.

Line 211: I would recommend saying average than bulk, as "average" best reflect what you present here and not a bulk analysis This figure is referred in Table 2 above, but I struggle to understand how the linear relationship was calculated.
We have added the word "calculated" to the "bulk" values. We are also sure to include quotation marks to denote that they are not a true bulk measurement. We prefer to keep this terminology as the calculated "bulk" data represent one of the possible sampling methods that has been used in

other works (Samperiz et al., 2020). We have also changed the reference from Table 2 as it was incorrectly pointing to this figure, we meant instead for it to reference Figure 5.

Line 251: Delete this "infinitesimally", it is an exaggeration! Just say with 5% aragonite.
Done.

Line 256: Delete this "for this discussion"
Done.

Line 257: Why O'Neil 1969 is chosen as there are many equilibrium equations too, explain your rationale in choosing their equation.
This equation was chosen because it was appropriate for our coral specimens (established for calcite-water fractionation at low temperatures) and it allowed our calculations to be directly comparable to those of Samperiz et al. (2020). Part of our discussion includes our data presented as an offset from isotopic equilibrium with seawater. We compare these quantities to the Samperiz et al. (2020) compilation (Figure 6), and therefore we prefer to use the same equations for a comparable discussion. If a different equilibrium equation was used for our calculation (e.g., Kim and O'Niel, 1997), the comparison to the Samperiz et al. (2020) compilation would not be accurate. We have added a statement to the text to clarify our choice and edited this paragraph slightly to clearly state the choice for each equilibrium equation.

Line 274: If you say strong, what is the p-values?
We have added the p-value (p<0.001) a few lines down with the other linear regression parameters and also in the caption for Figure 5.

Line 278: This is only true as you used one equilibrium equation, have you tested with different equation and was there any differences? This is the reason why I asked above for you to provide a rationale for why you chose O'Neil equation.
The statement to which the editor is referring remains true for different equilibrium equations. We describe the observation of coral isotope ratios from the center of the disc being closer to equilibrium than the values from the outer sections. The two most appropriate calcite-water equilibrium fractionation equations for this work are from O'Neil et al. (1969) and Kim and O'Neil (1997). These works conducted calcite precipitation experiments over a range of temperatures including 0°C, which is closest to our study site with a temperature of ~-0.1°C. For this manuscript, we calculated $\delta^{18}O$ equilibrium for calcite at 3.66 (+/- 0.06) ‰. For this review, we also calculated calcite $\delta^{18}O$ equilibrium using the equation from Kim and O'Neil (1997) to be 3.38 (+/- 0.06) ‰. This value is very close and would only minimally shift the equilibrium point (see yellow marker in figure below). We also used this calibration to remain consistent for comparing to equilibrium calculations in the literature.

*"**Figure 5**: Linear regressions of $\delta^{18}O$ vs $\delta^{13}C$ values for E. fissurata compared to aragonitic scleractinian and calcitic bamboo corals. Colors of circles (EA-11) and squares (EA-12) correspond to distance from the coral center, see color bar at the right. Calculated seawater equilibrium value for E. fissurata is also shown as a black rounded square (uncertainty is smaller than the square). The dashed black line represents the line of best fit for the isotopic values measured here ($\delta^{13}C$ = 2.88 (±0.14) * $\delta^{18}O$-10.94 (±0.22); p < 0.001). Linear regressions for Desmophyllum spp. are reported by Adkins et al. (2003) and include Desmophyllum sp. (purple line with squares) and D. dianthus (all other Desmophyllum lines). The dashed lines with shapes have corresponding equilibria displayed*

*(matching shape with error bars in upper right corner). The Bathypsammia tintinnabulum was reported by Emiliani et al. (1978) and Bamboo coral data are from Hill et al. (2011). Lines for external data are not extrapolated beyond the range of reported δ¹⁸O values. The slope of the linear regression produced in this study is similar to those reported for other deep-sea corals, with a similar decrease in both isotopic ratios from equilibrium. The measured values here that are closest to equilibrium are those toward the center of each coral disc."*

[Figure]

Line 294: Just curious, did they use the same equation as you used in this study? I may have missed this before.
Yes, when we compare our data to the published works, we are sure to use calcitic specimens for which equilibrium values were similarly calculated.

Line 319: Fig. 4is this the correct figure ref.? I don't see that strong variations in there, while accounting for the wide isotopic range.
We thank the editor for this catch, it is very difficult to see the values we mention on Figure 4, so we have changed the reference to direct the reader to Table 2 where the values are listed.

Line 323: See comments above, use average.
We have responded above to a similar comment.

Line 337: are (delete "were determined to be")
Done.

Line 337: Deletion of this last portion of this sentence.
Done.

Line 345: So when you say scanned, is this a different method than XRD?
No, this is still an XRD method.

Line 373: Refer to the suppl. figure if this is what you meant?
This sentence was meant to summarize the findings of Black and Andrus (2009), and we see that it was unclear. We have edited this sentence to clarify.

Line 376: Refer to suppl. and if possible, show the other figures of EA-22 and 23 instead of EA 24 alone, and indicate whereabout in the sample this image was taken. The scale for SEM and the whole sample is different.
We added a reference to the supplement, but unfortunately, we do not have images of EA-22 or EA-23. We have noted in the Figure captions of S4 and S5 that the scales are different (the scale bars are in the images).

Line 522: This method, if used, should be described further up (or some writing here need to be rephrased to make the meaning more accurate.
We agree that the wording was not clear about future applications of this method, so this section has been edited.

Line 543: See comments above about "bulk".
Again, we think that using the term "bulk" allows the reader to easily compare to other works wherein a true bulk measurement was made by drilling (e.g., King et al., 2018; Samperiz et al., 2020). We have placed quotation marks around each use of "bulk", and added the word "calculated" in front of it when the value was calculated to differentiate that it was not a true bulk measurement.

---

## Author Response (AR5)

We are grateful to the original reviewers for providing useful feedback on this manuscript again and thank them for their commitment to improving this work for publication. We have provided our responses to their comments below in blue.

**Associate editor decision: Publish subject to technical corrections**

by Ny Riavo G. Voarintsoa

**Public justification (visible to the public if the article is accepted and published)**:
Dear authors,

I have sent your revised manuscript to the original reviewers, who graciously agreed to review the revised documents you provided.
Both responses are positive towards the acceptance, but until then, please note that there are some technical corrections that were suggested by the reviewers. These are copied below:
* * *
In this reviewed version, the authors have included mineralogical data from one of the samples analyzed for stable isotopes, rather than from other samples collected at the same location. The main change (which is not minor) is that previous samples (EA-13 and EA-14) analyzed via XRD had >20 % aragonite, while current sample (EA-11b and EA-11d) have <5% aragonite. This update does change slightly the conclusions that can be drawn from this study, but this is not reflected in the manuscript in the current state. Below I suggest some minor changes regarding this issue, and some others. Line numbers are referring to the corrected version of the manuscript (not the one with tracked changes).

In section 4.2.3. Calcite versus aragonite mineralogy, the authors correctly identify the variable mineralogy present in E. fissurata and make a point that "mineralogy must be considered as an important variable […] use as a paleoceanographic archive. This should be included also in section 4.4 Considerations for paleoceanographic reconstructions, but there is no mention of mineralogy. We have added a statement in the beginning of section 4.4 to clarify that the discussion of paleotemperatures from these corals is based on the calcite mineralogy determined in this study. We have added that future projects should take careful consideration of the mineralogy of corals as it will affect interpretations of isotopic records.

Furthermore, on Line 493 the authors state "we recommend sampling of the white center using more spacially precise[…].". This would only be a viable sampling method for those samples that were >95% (Like EA-11b and EA-11d) but could be problematic for other samples that present mixed mineralogy. The authors have proved that even within the same location, mineralogy of samples can be variable, therefore, it is important to remark that testing for mineralogy is essential before proceeding with any geochemical measurement. We have added language in this paragraph to emphasize the importance of coral mineralogy, and that our interpretations apply to instances where coral mineralogy is spatially consistent.

Similarly, in the Conclusions, Line 526 "Thus, we recommend sampling this taxon along the center, white region where the carbonate geochemical record is closest to seawater equilibrium and environmental isotopic signal", there is absolutely no mention of the need to confirm that the

specimens chosen for paleoceanographic reconstructions need to be tested for mineralogy. The same applied with the last sentence of the abstract.
We have edited the conclusions and abstract to include the importance of determining mineralogy before applying our prescribed sampling plan.

The fact that sample EA-11 is >95% calcite and shows higher isotopic composition towards the center is indicating that these specimens might be following a different calcification process than other stylasterid corals (as said on the text), but this is contingent on mineralogy. Therefore, testing for mineralogy is imperative and as such must be included in the abstract and conclusions.
We agree, and appreciate this reviewer noticing the small oversight. We have added appropriate language to emphasize this conclusion.

Line 38 to 41: I find confusing the description of slow calcification, and then biological calcification as the opposite. I am not sure the authors are referring to abiotic calcification when talking about slow calcification, or about slow growth that allows calcium carbonate to grow in isotopic equilibrium with seawater (in this case it would still be biologically mediated growth, but at a slower pace?). Maybe the authors can rewrite for clarity.
Done

Line 290: "Erinna". I would describe it as E. antarctica, or the aragonitic E. antarctica as it is the way this specimen was referred to earlier.
This line was referring to two species within the *Errina* genus, but we have edited for clarity and stated *E. dabneyi* and *E. antarctica*.

Line 303: "This contrasts the increase we observe in our corals". Missing "with"?
Done

Section 4.2.1. Organic contribution to outer skeletal portion. This is a nice addition.
Thank you

Line 441: "The stylasterid Errina (Errina) labiata [...]". Why that double Errina and one within parenthesis?
The double *Errina* was because that was the name of the subgenus. Upon rechecking the name, it was listed as no longer accepted, so we have edited this line to include the updated, accepted name, *Inferiolabiata labiata*.

Line 487: "but (a) closer approach is possible".
Done

Line 512: "E. fissurata, a predominantly calcitic taxon". I disagree with this statement given evidence of XRD analysis of previous samples. Also, it is not in line with what it is stated in Line 366.
We agree and have removed this statement.
--
In addition to these comments, please ensure that the contribution is free of errors and typos as

possible (e.g., Figure S4: there is a typo, it should be "bottom"; Table S4: "Temperatures" is not countable and should not be in plural)

Done